# EpiLPS: A fast and flexible Bayesian tool for estimation of the time-varying reproduction number

**Oswaldo Gressani** [1] *, **Jacco Wallinga** [2,3], **Christian L. Althaus** [4], **Niel Hens** [1,5], **Christel Faes** [1]

**1** Interuniversity Institute for Biostatistics and statistical Bioinformatics (I-BioStat), Data Science Institute, Hasselt University, Hasselt, Belgium, **2** Centre for Infectious Disease Control, National Institute for Public Health and the Environment, Bilthoven, The Netherlands, **3** Department of Biomedical Data Sciences, Leiden University Medical Centre, Leiden, The Netherlands, **4** Institute of Social and Preventive Medicine, University of Bern, Bern, Switzerland, **5** Centre for Health Economics Research and Modelling Infectious Diseases, Vaxinfectio, University of Antwerp, Antwerp, Belgium

* oswaldo.gressani@uhasselt.be

**Data Availability Statement:** Simulation results and real data applications in this paper can be fully reproduced with the code available on the GitHub repository https://github.com/oswaldogressani/

## Abstract

In infectious disease epidemiology, the instantaneous reproduction number $\mathcal{R}_t$ is a time-varying parameter defined as the average number of secondary infections generated by an infected individual at time $t$. It is therefore a crucial epidemiological statistic that assists public health decision makers in the management of an epidemic. We present a new Bayesian tool (EpiLPS) for robust estimation of the time-varying reproduction number. The proposed methodology smooths the epidemic curve and allows to obtain (approximate) point estimates and credible intervals of $\mathcal{R}_t$ by employing the renewal equation, using Bayesian P-splines coupled with Laplace approximations of the conditional posterior of the spline vector. Two alternative approaches for inference are presented: (1) an approach based on a maximum a posteriori argument for the model hyperparameters, delivering estimates of $\mathcal{R}_t$ in only a few seconds; and (2) an approach based on a Markov chain Monte Carlo (MCMC) scheme with underlying Langevin dynamics for efficient sampling of the posterior target distribution. Case counts per unit of time are assumed to follow a negative binomial distribution to account for potential overdispersion in the data that would not be captured by a classic Poisson model. Furthermore, after smoothing the epidemic curve, a "plug-in" estimate of the reproduction number can be obtained from the renewal equation yielding a closed form expression of $\mathcal{R}_t$ as a function of the spline parameters. The approach is extremely fast and free of arbitrary smoothing assumptions. EpiLPS is applied on data of SARS-CoV-1 in Hong-Kong (2003), influenza A H1N1 (2009) in the USA and on the SARS-CoV-2 pandemic (2020-2021) for Belgium, Portugal, Denmark and France.

## Author summary

The instantaneous reproduction number $\mathcal{R}_t$ is a key statistic that provides important insights into an epidemic outbreak as it informs about the average number of secondary

EpiLPS-ArticleCode based on the EpiLPS package version 1.0.6 available on CRAN (https://cran.r-project.org/package=EpiLPS).

**Funding:** This project is funded by the European Union's Research and Innovation Action (https://cordis.europa.eu/project/id/101003688) under the H2020 work programme, EpiPose grant number 101003688. The funders had no role in study design, data collection and analysis, decision to publish, or preparation of the manuscript.

**Competing interests:** The authors have declared that no competing interests exist.

infections engendered by an infectious agent. We present a flexible Bayesian approach called EpiLPS (**Epi**demiological modeling with **L**aplacian-**P**-**S**plines) for efficient estimation of the epidemic curve and $\mathcal{R}_t$ based on daily case count data and the serial interval distribution. Computational speed and absence of arbitrary assumptions on smoothing makes EpiLPS an interesting tool for estimation of the reproduction number. Our methodology is validated through different simulation scenarios by using the associated R software package (https://cran.r-project.org/package=EpiLPS). We also demonstrate the use of EpiLPS on real data from two historical outbreaks and on the SARS-CoV-2 pandemic.

This is a *PLOS Computational Biology* Methods paper.

## Introduction

The instantaneous reproduction number $\mathcal{R}_t$ is a time-varying parameter defined as the average number of secondary cases generated by an infectious individual at time *t*. During epidemic outbreaks, $\mathcal{R}_t$ provides a snapshot (often on a daily basis) that quantifies the extent to which a given infectious disease transmits in a population and is therefore an important tool that assists governmental organizations in the management of a public health crisis. The reproduction number is also a good proxy for measuring the real-time growth phase of an epidemic and as such, constitutes a key signal about the transmission potential of the outbreak and the required control effort. For this reason, having a robust, accurate and timely estimator of $\mathcal{R}_t$ is a crucial matter that has attracted considerable interest in developing new statistical approaches during the last two decades as summarized in [1]. The paper of [2] compares several methods for estimating $\mathcal{R}_t$ and gives clear insights about the main challenges and obstacles that have to be faced. They recommend the method of [3] and its associated EpiEstim package [4] as an appropriate and accurate tool for near real-time estimation of the instantaneous reproduction number. Another recent approach is proposed in [5], where a recursive Bayesian smoother based on Kalman filtering is used to derive a robust estimate of $\mathcal{R}_t$ in periods of low incidence. The EpiNow2 package [6] also provides interesting extensions and implementations of current best practices for precise estimation and forecast of the reproduction number using a Bayesian latent variable framework. Spline based approaches have shown to be a useful tool for flexible modeling of the reproduction number. [7] use penalized radial splines for estimating $\mathcal{R}_t$ under a Bayesian setting with misreported data and [8] accelerated the computational implementation by replacing the Markov chain Monte Carlo (MCMC) scheme with Laplace approximations. From a frequentist perspective, [9] uses truncated polynomials and radial basis splines to model the series of new infections and a derivative thereof as a candidate estimator for the reproduction number.

In this article, we propose a new Bayesian approach termed "EpiLPS" for estimating $\mathcal{R}_t$ based on case incidence data and the serial interval (SI) distribution (the time elapsed between the onset of symptoms in an infector and the onset of symptoms in the secondary cases generated by that infector). Our estimator of $\mathcal{R}_t$ is based on epidemic renewal equations [10, 11] and Laplacian-P-splines smoothing of the mean number of incidence cases. Time series of new cases by day of reporting (or day of symptom onset) are assumed to follow a negative binomial distribution to account for potential excess variability as frequently encountered in epidemiological count data. Algorithms related to Laplace approximations and evaluations of

B-spline bases are coded in C++ and embedded in the R language through the Rcpp package [12], making computational speed another key strength of EpiLPS as $\mathcal{R}_t$ can be estimated in seconds. In addition, EpiLPS can also be used to obtain a smoothed estimate of the epidemic curve that can be of potential interest to further visualize an epidemic outbreak.

The proposed Bayesian methodology is based on a latent Gaussian model for the B-spline amplitudes and opens up two possible paths for inference. The first is called LPSMAP, a fully sampling-free approach based on Laplace approximations to the conditional posterior of B-spline coefficients. The hyperparameter vector is fixed at its maximum *a posteriori* and credible intervals of $\mathcal{R}_t$ are computed via the "delta" method. The second path is called LPSMALA and is a MCMC approach based on the Langevin diffusion for efficient exploration of the posterior distribution of latent variables. The latter approach is computationally heavier than LPSMAP but has the merit of taking into account the uncertainty surrounding the hyperparameters. The underlying Metropolis-within-Gibbs structure keeps the practical implementation to a fairly simple level and the computational cost is reasonable even for long chains.

Compared to existing methods, EpiLPS resembles EpiEstim from a methodological point of view in the sense that $\mathcal{R}_t$ is estimated from incidence time series and a serial interval distribution, yet the two approaches fundamentally differ in many aspects. First, the methodology of [3] assumes that incidence at time *t* is Poisson distributed, while EpiLPS assumes a negative binomial model. Second, as our approach uses penalized spline based approximations, prior specifications are imposed on the roughness penalty parameter and not directly on $\mathcal{R}_t$ as in EpiEstim. Third and most importantly, EpiLPS is free of any sliding window specification, while EpiEstim relies on a user-defined time window. This subjective time window choice is the key driving force that determines how smooth the estimated $\mathcal{R}_t$ trajectory will be. In EpiLPS, the optimal amount of smoothing is data-driven and objectively estimated (through the penalty parameter) within the Bayesian model. An R package for EpiLPS has been developed and is available at https://cran.r-project.org/package=EpiLPS. The software also allows to compute the Cori et al. (2013) [3] estimate of $\mathcal{R}_t$ for the sake of comparison.

The manuscript is organized as follows. We first present the Laplacian-P-splines model for smoothing count data and show how the Laplace approximation applies to the conditional posterior of the B-spline amplitudes and also derive the (approximate) posterior of the hyperparameter vector to be optimized. This yields the maximum *a posteriori* (MAP) estimate of the spline vector via Laplacian-P-splines (LPSMAP). We then use LPSMAP to propose a "plug-in" estimate of $\mathcal{R}_t$ based on renewal equations and proceed to the computation of credible intervals. An alternative path for estimation of $\mathcal{R}_t$ based on MCMC is also presented. The latter approach uses Langevin dynamics for efficient sampling of the target posterior distribution and is termed LPSMALA for "Laplacian-P-splines with a Metropolis-adjusted Langevin algorithm". Next, we assess the performance of EpiLPS in various simulation scenarios and make comparisons with EpiEstim. Finally, we apply EpiLPS to real world epidemic outbreaks before concluding with a discussion.

## Methods

### Negative binomial model for case incidence data

Let $\mathcal{D} = \{y_t, t = 1, \dots, T\}$ be a time series of counts during an epidemic of $T$ days with $y_t \in \mathbb{N}$ (set of non-negative integers) denoting the number of cases by reporting date or by date of symptom onset. We assume that the number of cases on day *t* follows a negative binomial distribution $y_t \sim \mathrm{NegBin}(\mu(t), \rho)$, with $\mu(t), \rho \in \mathbb{R}_+^* := \{x \in \mathbb{R} | x > 0\}$ and probability mass

function (see e.g. [13, 14]):

$$p(y_t | \mu(t), \rho) = \frac{\Gamma(y_t + \rho)}{\Gamma(y_t + 1)\Gamma(\rho)} \left( \frac{\mu(t)}{\mu(t) + \rho} \right)^{y_t} \left( \frac{\rho}{\rho + \mu(t)} \right)^{\rho}, \tag{1}$$

where $\Gamma(\cdot)$ is the gamma function. The above parameterization is frequently encountered in epidemiology [15] and yields a mean $\mathbb{E}(y_t) = \mu(t)$ and variance $\mathbb{V}(y_t) = \mu(t) + \mu(t)^2/\rho$, so that $\rho$ is the parameter responsible for overdispersion (variance larger than the mean) that is absent in a Poisson setting. In the limiting case $\lim_{\rho \to +\infty} \mathbb{V}(y_t) = \mu(t) = \mathbb{E}(y_t)$ and we recover the mean-variance equality of the Poisson model. The key argument in favor of a negative binomial distribution is thus its ability to capture the often encountered feature of overdispersion present in infectious disease count data [16]. We assume that $\mu(t)$ evolves smoothly over the time course of the epidemic and model it with cubic B-splines [17]:

$$\log(\mu(t)) = \sum_{k=1}^{K} \theta_k b_k(t) = \boldsymbol{\theta}^\top b(t), \tag{2}$$

where $\boldsymbol{\theta} = (\theta_1, \ldots, \theta_K)^\top$ is the vector of B-spline amplitudes to be estimated and $b(\cdot) = (b_1(\cdot), \ldots, b_K(\cdot))^\top$ is a cubic B-spline basis defined on the domain $\mathcal{T} = [r_l, T]$, where $r_l$ is a lower bound on the time axis, typically the first day of the epidemic (i.e. $r_l = 1$). The philosophy behind P-splines consists in specifying a "large" number $K$ of basis functions together with a discrete roughness penalty $\lambda \boldsymbol{\theta}^\top P \boldsymbol{\theta}$ as a counterforce to the induced flexibility of the fit. The parameter $\lambda > 0$ acts as a tuning parameter calibrating the "degree" of smoothness and $P = D_r^\top D_r + \varepsilon I_K$ is a penalty matrix built from $r$th order difference matrices $D_r$ of dimension $(K - r) \times K$ perturbed by an $\varepsilon$-multiple (here $\varepsilon = 10^{-6}$) of the $K$-dimensional identity matrix $I_K$ to ensure full rankedness. There are several attractive reasons to use P-splines for smoothing the epidemic curve and $\mathcal{R}_t$. First, as the P-splines setting specifies an abundant number of B-spline basis functions coupled with a penalty on the spline coefficients to control for overfitting, the resulting $\mu(t)$ fit is smooth and estimates can be obtained for any $t$ on the continuous time domain. Second, even if the number $K$ of B-splines is free to choose, the shape of the fitted $\mathcal{R}_t$ curve is actually regulated by the smoothing parameter $\lambda$ and hence only negligibly affected by the arbitrary choice of $K$, provided it is large enough [18]. Third, the intrinsic sparseness of $P$ and of the B-spline basis matrix is computationally appealing as it softens the algorithmic implementation and yields numerically stable routines [19, 20]. Another key advantage of P-splines smoothers is their natural formulation in a Bayesian framework by translating difference penalties on contiguous B-spline coefficients into Gaussian random walk smoothness priors [21]. Following the latter reference, we impose a Gaussian prior on the vector of spline coefficients $\boldsymbol{\theta} | \lambda \sim \mathcal{N}_{\dim(\boldsymbol{\theta})}(0, Q_\lambda^{-1})$, with precision matrix $Q_\lambda = \lambda P$. For full Bayesian inference, the following priors are imposed on the model hyperparameters. Following [22], a robust Gamma prior is specified for the roughness penalty parameter $\lambda | \delta \sim \mathcal{G}(\phi/2, (\phi\delta)/2)$, where $\mathcal{G}(a, b)$ is a Gamma distribution with mean $a/b$ and variance $a/b^2$, $\phi = 2$ and $\delta$ is an additional dispersion parameter with hyperprior $\delta \sim \mathcal{G}(a_\delta = 10, b_\delta = 10)$. This prior specification favors "small" $\lambda$ values and translates the belief that a wiggly $\mathcal{R}_t$ fit is more inclined to arise during the epidemic period as opposed to an oversmoothed fit. Finally, the following uninformative prior is imposed on the overdispersion parameter $\rho \sim \mathcal{G}(a_\rho = 0.0001, b_\rho = 0.0001)$. Let $\boldsymbol{\eta} :=$

$(\lambda, \rho)^\top$ denote the vector of hyperparameters. The full Bayesian model is thus:

$$
\begin{aligned}
y_t|\mu(t), \rho &\sim \text{NegBin}(\mu(t), \rho), \\
\log(\mu(t)) &= \boldsymbol{\theta}^\top b(t), \\
\boldsymbol{\theta}|\lambda &\sim \mathcal{N}_{\dim(\boldsymbol{\theta})}(0, Q_\lambda^{-1}), \\
\lambda|\delta &\sim \mathcal{G}(\phi/2, (\phi\delta)/2), \\
\delta &\sim \mathcal{G}(a_\delta, b_\delta), \\
\rho &\sim \mathcal{G}(a_\rho, b_\rho).
\end{aligned}
$$

## Laplace approximation to the conditional posterior of $\boldsymbol{\theta}$

The Laplace approximation has two key roles in the proposed EpiLPS methodology. First, it determines the approximating distribution to the (conditional) posterior of the spline vector $\boldsymbol{\theta}$ that will be used to estimate the average incidence of cases at time $t$, i.e. $\mathbb{E}(y_t)$ and hence $\mathcal{R}_t$ via the renewal equation. Second, the variance-covariance matrix of the Laplace approximation is used to quantify the uncertainty of the instantaneous reproduction number through a "delta" method in LPSMAP and is also introduced in the proposal distribution of the LPSMALA algorithm to form the skeleton of the correlation structure for the spline components. The synergy between Laplace approximations and P-splines has already been shown to be very effective for modeling count data (see for instance [23], in the context of generalized additive models). The log-likelihood for the negative binomial model is given by:

$$
\ell(\boldsymbol{\theta}, \rho; \mathcal{D}) \dot{=} \sum_{t=1}^{T} \{g(y_t, \rho) + y_t \boldsymbol{\theta}^\top b(t) + \rho \log(\rho) - (y_t + \rho) \log(\exp(\boldsymbol{\theta}^\top b(t)) + \rho)\}, \tag{3}
$$

with $g(y_t, \rho) = \log \Gamma(y_t + \rho) - \log \Gamma(\rho)$ and $\dot{=}$ denoting equality up to an additive constant. The gradient of the log-likelihood with respect to the spline coefficients is:

$$
\nabla_{\boldsymbol{\theta}} \ell(\boldsymbol{\theta}, \rho; \mathcal{D}) = \left( \frac{\partial \ell(\boldsymbol{\theta}, \rho; \mathcal{D})}{\partial \theta_1}, \ldots, \frac{\partial \ell(\boldsymbol{\theta}, \rho; \mathcal{D})}{\partial \theta_K} \right)^\top,
$$

where:

$$
\frac{\partial \ell(\boldsymbol{\theta}, \rho; \mathcal{D})}{\partial \theta_k} = \sum_{t=1}^{T} y_t b_k(t) - \sum_{t=1}^{T} \frac{(y_t + \rho) \exp(\boldsymbol{\theta}^\top b(t))}{(\exp(\boldsymbol{\theta}^\top b(t)) + \rho)} b_k(t), \quad k = 1, \ldots, K.
$$

The Hessian of the log-likelihood with respect to the B-spline amplitudes is:

$$
\nabla_{\boldsymbol{\theta}}^2 \ell(\boldsymbol{\theta}, \rho; \mathcal{D}) = \begin{pmatrix} \dfrac{\partial^2 \ell(\boldsymbol{\theta}, \rho; \mathcal{D})}{\partial \theta_1^2} & \cdots & \dfrac{\partial^2 \ell(\boldsymbol{\theta}, \rho; \mathcal{D})}{\partial \theta_1 \partial \theta_K} \\ \vdots & \ddots & \vdots \\ \dfrac{\partial^2 \ell(\boldsymbol{\theta}, \rho; \mathcal{D})}{\partial \theta_K \partial \theta_1} & \cdots & \dfrac{\partial^2 \ell(\boldsymbol{\theta}, \rho; \mathcal{D})}{\partial \theta_K^2} \end{pmatrix},
$$

with entries:

$$
\frac{\partial^2 \ell(\boldsymbol{\theta}, \rho; \mathcal{D})}{\partial \theta_k \partial \theta_l} = -\sum_{t=1}^{T} \rho(y_t + \rho) \frac{\exp(\boldsymbol{\theta}^\top b(t))}{(\exp(\boldsymbol{\theta}^\top b(t)) + \rho)^2} b_k(t) b_l(t), \quad k, l = 1, \ldots, K.
$$

Using Bayes' rule, the conditional posterior of $\boldsymbol{\theta}$ for a given $\boldsymbol{\eta}$ is:

$$
\begin{aligned}
p(\boldsymbol{\theta}|\boldsymbol{\eta}, \mathcal{D}) &\propto \mathcal{L}(\boldsymbol{\theta}, \rho; \mathcal{D})p(\boldsymbol{\theta}|\lambda) \\
&\propto \exp\left(\ell(\boldsymbol{\theta}, \rho; \mathcal{D}) - \frac{\lambda}{2}\boldsymbol{\theta}^{\top}P\boldsymbol{\theta}\right),
\end{aligned}
\tag{4}
$$

where $\mathcal{L}(\boldsymbol{\theta}, \rho; \mathcal{D})$ denotes the likelihood function. The gradient and Hessian of the log-likelihood (3) can be used to compute the gradient and Hessian of the (log-)conditional posterior (4), namely:

$$
\nabla_{\boldsymbol{\theta}} \log p(\boldsymbol{\theta}|\boldsymbol{\eta}, \mathcal{D}) = \nabla_{\boldsymbol{\theta}}\ell(\boldsymbol{\theta}, \rho; \mathcal{D}) - \lambda P\boldsymbol{\theta},
$$
$$
\nabla^2_{\boldsymbol{\theta}} \log p(\boldsymbol{\theta}|\boldsymbol{\eta}, \mathcal{D}) = \nabla^2_{\boldsymbol{\theta}}\ell(\boldsymbol{\theta}, \rho; \mathcal{D}) - \lambda P.
$$

The above two equations will be used iteratively in a Newton-Raphson algorithm to obtain the Laplace approximation to the conditional posterior of $\boldsymbol{\theta}$:

$$
\widetilde{p}_G(\boldsymbol{\theta}|\boldsymbol{\eta}, \mathcal{D}) = \mathcal{N}_{\dim(\boldsymbol{\theta})}(\boldsymbol{\theta}^*(\boldsymbol{\eta}), \Sigma^*(\boldsymbol{\eta})),
\tag{5}
$$

where $\boldsymbol{\theta}^*(\boldsymbol{\eta})$ and $\Sigma^*(\boldsymbol{\eta})$ is the mode and variance-covariance respectively after convergence of the Newton-Raphson algorithm. The latter two quantities are functions of the hyperparameter vector $\boldsymbol{\eta}$. An intuitive choice for $\boldsymbol{\eta}$ is to fix it at its maximum *a posteriori*. This is the option retained here, although it is also possible to work with a grid-based approach [23, 24].

## Hyperparameter optimization

The hyperparameter vector $\boldsymbol{\eta} = (\lambda, \rho)^{\top}$ will be calibrated by posterior optimization. Following [25] and [24], the hyperparameter vector can be approximated as follows:

$$
\widetilde{p}(\boldsymbol{\eta}, \delta|\mathcal{D}) \propto \frac{\mathcal{L}(\boldsymbol{\theta}, \rho; \mathcal{D})p(\boldsymbol{\theta}|\lambda)p(\lambda|\delta)p(\delta)p(\rho)}{\widetilde{p}_G(\boldsymbol{\theta}|\boldsymbol{\eta}, \mathcal{D})}\bigg|_{\boldsymbol{\theta}=\boldsymbol{\theta}^*(\boldsymbol{\eta})}.
\tag{6}
$$

Approximation (6) can be written extensively as:

$$
\begin{aligned}
\widetilde{p}(\boldsymbol{\eta}, \delta|\mathcal{D}) &\propto \lambda^{\frac{K+\phi}{2}-1}\delta^{\frac{\phi}{2}+a_\delta-1}\exp\left(-\delta\left(\frac{\phi\lambda}{2} + b_\delta\right)\right)\rho^{a_\rho-1} \\
&\times |\Sigma^*(\boldsymbol{\eta})|^{\frac{1}{2}}\exp\left(\ell(\boldsymbol{\theta}^*(\boldsymbol{\eta}), \rho; \mathcal{D}) - \frac{\lambda}{2}\boldsymbol{\theta}^{*\top}(\boldsymbol{\eta})P\boldsymbol{\theta}^*(\boldsymbol{\eta}) - b_\rho\rho\right),
\end{aligned}
$$

where the $K/2$ power of $\lambda$ comes from the determinant $|Q_\lambda^{-1}|^{-1/2} = |\lambda P|^{1/2} \propto \lambda^{K/2}$. As $\delta^{\frac{\phi}{2}+a_\delta-1}\exp\left(-\delta\left(\frac{\phi\lambda}{2} + b_\delta\right)\right)$ is the kernel of a Gamma distribution for the dispersion parameter $\delta$, the following integral can be analytically solved:

$$
\begin{aligned}
\int_0^{+\infty} \widetilde{p}(\boldsymbol{\eta}, \delta|\mathcal{D})\ d\delta &= \widetilde{p}(\boldsymbol{\eta}|\mathcal{D}) \\
&\propto \lambda^{\frac{K+\phi}{2}-1}\left(\frac{\phi\lambda}{2} + b_\delta\right)^{-\left(\frac{\phi}{2}+a_\delta\right)}\rho^{a_\rho-1} \\
&\times |\Sigma^*(\boldsymbol{\eta})|^{\frac{1}{2}}\exp\left(\ell(\boldsymbol{\theta}^*(\boldsymbol{\eta}), \rho; \mathcal{D}) - \frac{\lambda}{2}\boldsymbol{\theta}^{*\top}(\boldsymbol{\eta})P\boldsymbol{\theta}^*(\boldsymbol{\eta}) - b_\rho\rho\right).
\end{aligned}
$$

Using the transformation of variables (ensuring numerical stability during optimization) $w = \log(\rho)$, $v = \log(\lambda)$, one can show that $\widetilde{p}(\boldsymbol{\eta}|\mathcal{D})$ can be written as follows after using the

multivariate transformation method:

$$\widetilde{p}(\widetilde{\boldsymbol{\eta}}|\mathcal{D}) \quad \propto \quad \exp(v)^{\frac{K+\phi}{2}}\left(\frac{\phi\exp(v)}{2}+b_\delta\right)^{-\left(\frac{\phi}{2}+a_\delta\right)}\exp(w)^{a_\rho}$$

$$\times|\Sigma^*(\widetilde{\boldsymbol{\eta}})|^{\frac{1}{2}}\exp\left(\ell(\boldsymbol{\theta}^*(\widetilde{\boldsymbol{\eta}}),\exp(w);\mathcal{D})-\frac{\exp(v)}{2}\boldsymbol{\theta}^{*\top}(\widetilde{\boldsymbol{\eta}})P\boldsymbol{\theta}^*(\widetilde{\boldsymbol{\eta}})-b_\rho\exp(w)\right),$$

where $\widetilde{\boldsymbol{\eta}} = (w,v)^\top$. The approximated log-posterior becomes:

$$\log\widetilde{p}(\widetilde{\boldsymbol{\eta}}|\mathcal{D}) \doteq 0.5\log|\Sigma^*(\widetilde{\boldsymbol{\eta}})| + 0.5(K+\phi)v + a_\rho w - (0.5\phi+a_\delta)\log\left(0.5\phi\exp(v)+b_\delta\right)$$
$$+\ell(\boldsymbol{\theta}^*(\widetilde{\boldsymbol{\eta}}),\exp(w);\mathcal{D}) - 0.5\exp(v)\boldsymbol{\theta}^{*\top}(\widetilde{\boldsymbol{\eta}})P\boldsymbol{\theta}^*(\widetilde{\boldsymbol{\eta}}) - b_\rho\exp(w). \tag{7}$$

Eq (7) is numerically optimized and yields $\widetilde{\boldsymbol{\eta}}^* = \text{argmax}_{\widetilde{\boldsymbol{\eta}}}\log\widetilde{p}(\widetilde{\boldsymbol{\eta}}|\mathcal{D})$. Plugging the latter vector into the Laplace approximation (5), we obtain the estimate $\hat{\boldsymbol{\theta}} = \boldsymbol{\theta}^*(\widetilde{\boldsymbol{\eta}}^*)$ of the spline vector. The latter can be seen as a MAP estimate of $\boldsymbol{\theta}$. Thus, the approximated (conditional) posterior of the spline vector is:

$$\widetilde{p}_G(\boldsymbol{\theta}|\widetilde{\boldsymbol{\eta}}^*,\mathcal{D}) = \mathcal{N}_{\dim(\boldsymbol{\theta})}(\boldsymbol{\theta}^*(\widetilde{\boldsymbol{\eta}}^*),\Sigma^*(\widetilde{\boldsymbol{\eta}}^*)), \tag{8}$$

and can be used to construct credible intervals for functions that depend on $\boldsymbol{\theta}$, such as $\mathcal{R}_t$ as shown in the following section.

## Estimation of $\mathcal{R}_t$ with LPSMAP

**The renewal equation "plug-in" estimate.** In this section, we show how the negative binomial model for smoothing incidence counts can be used to estimate $\mathcal{R}_t$ through the renewal equation. Let $\boldsymbol{\varphi} = \{\varphi_1, \ldots, \varphi_k\}$ be a known $k$-dimensional vector representing the serial interval (SI) distribution, where $\varphi_s$ is the probability that the SI is equal to $s$ day(s), i.e. $\varphi_s = \mathbb{P}(SI = s)$. We also assume $\sum_{s=1}^k \varphi_s = 1$ and $\mathbb{P}(SI \leq 0) = \mathbb{P}(SI > k) = 0$. The renewal model [10, 11] gives a mathematical statement of equality between the mean incidence of cases at time step $t$ and a product between the reproduction number $\mathcal{R}_t$ and a convolution involving antecedent cases and the serial interval distribution:

$$\mathbb{E}(y_t) = \mathcal{R}_t\Lambda_t, \tag{9}$$

where $\Lambda_t = \sum_{s=1}^{t-1}\varphi_s y_{t-s}$ denotes the number of circulating cases that contribute to active transmission, also known as total infectiousness at time $t$ [5]. Rearranging Eq (9) and taking the length $k$ of the serial interval into account, we obtain an equation with the instantaneous reproduction number on the left-hand side:

$$\mathcal{R}_t = \begin{cases} \mathbb{E}(y_t) & \text{for } t = 1, \\ \mathbb{E}(y_t)\left(\sum_{s=1}^{t-1}\varphi_s y_{t-s}\right)^{-1} & \text{for } 2 \leq t \leq k, \\ \mathbb{E}(y_t)\left(\sum_{s=1}^{k}\varphi_s y_{t-s}\right)^{-1} & \text{for } k < t \leq T. \end{cases} \tag{10}$$

Our Bayesian "plug-in" estimator of $\mathcal{R}_t$ at time step $t$ is obtained by replacing the average number of cases $\mathbb{E}(y_t) = \mu(t)$ by the estimated average $\hat{\mu}(t) = \exp(\hat{\boldsymbol{\theta}}^\top b(t))$ and by replacing

$y_{t-s}$ by $\hat{\mu}(t-s) = \exp(\hat{\boldsymbol{\theta}}^\top b(t-s))$:

$$
\hat{\mathcal{R}}_t = \begin{cases}
\exp(\hat{\boldsymbol{\theta}}^\top b(t)) & \text{for } t = 1, \\[2mm]
\exp(\hat{\boldsymbol{\theta}}^\top b(t))(\sum_{s=1}^{t-1} \varphi_s \exp(\hat{\boldsymbol{\theta}}^\top b(t-s)))^{-1} & \text{for } 2 \leq t \leq k, \\[2mm]
\exp(\hat{\boldsymbol{\theta}}^\top b(t))(\sum_{s=1}^{k} \varphi_s \exp(\hat{\boldsymbol{\theta}}^\top b(t-s)))^{-1} & \text{for } k < t \leq T.
\end{cases} \tag{11}
$$

Note that the MAP estimate of the overdispersion parameter affects the estimate $\hat{\mu}(t)$ via $\hat{\boldsymbol{\theta}}$. Using the indicator function $\mathbb{I}(\cdot)$, i.e. $\mathbb{I}(A) = 1$ if condition $A$ is true and $\mathbb{I}(A) = 0$ otherwise, the above estimator can be written in a single line:

$$
\begin{aligned}
\hat{\mathcal{R}}_t &= \exp(\hat{\boldsymbol{\theta}}^\top b(t))\left\{ \mathbb{I}(t=1) + \left(\sum_{s=1}^{t-1} \varphi_s \exp(\hat{\boldsymbol{\theta}}^\top b(t-s))\right)^{-1} \mathbb{I}(2 \leq t \leq k) \right. \\
&\quad \left. + \left(\sum_{s=1}^{k} \varphi_s \exp(\hat{\boldsymbol{\theta}}^\top b(t-s))\right)^{-1} \mathbb{I}(k < t \leq T) \right\}.
\end{aligned} \tag{12}
$$

**Credible intervals for $\mathcal{R}_t$.** Using the functional relationship between $\mathcal{R}_t$ and $\boldsymbol{\theta}$ as in Eq (12), the log of the instantaneous reproduction number can be written as:

$$
\begin{aligned}
\log \mathcal{R}_t &:= h(\boldsymbol{\theta}|t) \\
&= \boldsymbol{\theta}^\top b(t) + \log \zeta(\boldsymbol{\theta}),
\end{aligned}
$$

with

$$
\begin{aligned}
\zeta(\boldsymbol{\theta}) &= \mathbb{I}(t=1) + \left(\sum_{s=1}^{t-1} \varphi_s \exp(\boldsymbol{\theta}^\top b(t-s))\right)^{-1} \mathbb{I}(2 \leq t \leq k) \\
&\quad + \left(\sum_{s=1}^{k} \varphi_s \exp(\boldsymbol{\theta}^\top b(t-s))\right)^{-1} \mathbb{I}(k < t \leq T).
\end{aligned}
$$

Note that $h(\boldsymbol{\theta}|t)$ is seen here as a function of the spline vector $\boldsymbol{\theta}$ for a given time point $t$. A $(1 - \alpha) \times 100\%$ approximate credible interval for $\mathcal{R}_t$ is obtained via a "delta" method. Consider a first-order Taylor expansion of $h(\boldsymbol{\theta}|t)$ around $\boldsymbol{\theta}^*(\widetilde{\boldsymbol{\eta}}^*)$ (henceforth $\boldsymbol{\theta}^*$ for the sake of a light notation), the mean of the Laplace approximated posterior of the spline vector in (8):

$$
h(\boldsymbol{\theta}|t) \approx h(\boldsymbol{\theta}^*|t) + (\boldsymbol{\theta} - \boldsymbol{\theta}^*)^\top \nabla h(\boldsymbol{\theta}|t)|_{\boldsymbol{\theta}=\boldsymbol{\theta}^*}, \tag{13}
$$

where the $k$th entry of the gradient vector $\nabla h(\boldsymbol{\theta}|t) = (\partial h(\boldsymbol{\theta}|t)/\partial\theta_1, \ldots, \partial h(\boldsymbol{\theta}|t)/\partial\theta_K)^\top$ is:

$$
\begin{aligned}
\frac{\partial h(\boldsymbol{\theta}|t)}{\theta_k} &= b_k(t) + \zeta^{-1}(\boldsymbol{\theta})\frac{\partial \zeta(\boldsymbol{\theta})}{\partial \theta_k}. \\
\frac{\partial \zeta(\boldsymbol{\theta})}{\partial \theta_k} &= -\left(\sum_{s=1}^{t-1} \varphi_s \exp(\boldsymbol{\theta}^\top b(t-s))\right)^{-2} \sum_{s=1}^{t-1} \varphi_s \exp(\boldsymbol{\theta}^\top b(t-s))b_k(t-s)\mathbb{I}(2 \leq t \leq k) \\
&\quad - \left(\sum_{s=1}^{k} \varphi_s \exp(\boldsymbol{\theta}^\top b(t-s))\right)^{-2} \sum_{s=1}^{k} \varphi_s \exp(\boldsymbol{\theta}^\top b(t-s))b_k(t-s)\mathbb{I}(k < t \leq T).
\end{aligned}
$$

It follows that for $k = 1, \ldots, K$, we have:

$$
\frac{\partial h(\boldsymbol{\theta}|t)}{\theta_k} = b_k(t) + \Bigg\{ 0\mathbb{I}(t = 1)
$$

$$
- \left( \sum_{s=1}^{t-1} \varphi_s \exp(\boldsymbol{\theta}^\top b(t-s)) \right)^{-1} \sum_{s=1}^{t-1} \varphi_s \exp(\boldsymbol{\theta}^\top b(t-s)) b_k(t-s) \mathbb{I}(2 \leq t \leq k)
$$

$$
- \left( \sum_{s=1}^{k} \varphi_s \exp(\boldsymbol{\theta}^\top b(t-s)) \right)^{-1} \sum_{s=1}^{k} \varphi_s \exp(\boldsymbol{\theta}^\top b(t-s)) b_k(t-s) \mathbb{I}(k < t \leq T) \Bigg\}.
$$

The Taylor expansion in (13) is a linear combination of the vector $\boldsymbol{\theta}$ that is *a posteriori* (approximately) Gaussian due to the Laplace approximation. As the family of Gaussian distributions is closed under linear combinations, it follows that $h(\boldsymbol{\theta}|t)$ (and hence $\log \mathcal{R}_t$) is *a posteriori* also (approximately) Gaussian with mean $\mathbb{E}(h(\boldsymbol{\theta}|t)) \approx h(\boldsymbol{\theta}^*|t)$ and variance $\mathbb{V}(h(\boldsymbol{\theta}|t)) \approx \nabla^\top h(\boldsymbol{\theta}|t)|_{\boldsymbol{\theta}=\boldsymbol{\theta}^*} \Sigma^* \nabla h(\boldsymbol{\theta}|t)|_{\boldsymbol{\theta}=\boldsymbol{\theta}^*}$, where $\Sigma^* := \Sigma^*(\widetilde{\boldsymbol{\eta}}^*)$ is the covariance matrix of the Laplace approximation (8). This suggests to write:

$$
(\log \mathcal{R}_t | \mathcal{D}) \approx \mathcal{N}_1 (h(\boldsymbol{\theta}^*|t), \nabla^\top h(\boldsymbol{\theta}|t)|_{\boldsymbol{\theta}=\boldsymbol{\theta}^*} \Sigma^* \nabla h(\boldsymbol{\theta}|t)|_{\boldsymbol{\theta}=\boldsymbol{\theta}^*}). \tag{14}
$$

The accuracy of the variance approximation in (14) can be improved through a scaling of the covariance matrix $\Sigma^*$ by multiplying it with the scaling factor $\kappa_t^{\hat{\rho}} := (1 + \hat{\rho}^{-1}\hat{\mu}(t))^{-1}$, corresponding to the estimated mean-to-variance ratio $\mathbb{E}(y_t)/\mathbb{V}(y_t)$ at time step $t$ (see S2 Appendix). The (approximate) posterior distribution for $\mathcal{R}_t$ is thus given by $(\mathcal{R}_t|\mathcal{D}) \sim LogNorm(\mu_{\mathcal{R}_t}^*, \sigma_{\mathcal{R}_t}^{2*})$, i.e. a lognormal distribution with parameters $\mu_{\mathcal{R}_t}^* = h(\boldsymbol{\theta}^*|t)$ and $\sigma_{\mathcal{R}_t}^{2*} = \nabla^\top h(\boldsymbol{\theta}|t)|_{\boldsymbol{\theta}=\boldsymbol{\theta}^*} \kappa_t^{\hat{\rho}} \Sigma^* \nabla h(\boldsymbol{\theta}|t)|_{\boldsymbol{\theta}=\boldsymbol{\theta}^*}$. A quantile-based $(1-\alpha) \times 100\%$ approximate credible interval for $\mathcal{R}_t$ is thus $CI_{\mathcal{R}_t}^{1-\alpha} = \exp(\mu_{\mathcal{R}_t}^* \pm z_{\alpha/2}\sigma_{\mathcal{R}_t}^*)$, where $z_{\alpha/2}$ is the $\alpha/2$-upper quantile of a standard normal variate.

## Estimation of $\mathcal{R}_t$ with LPSMALA

In Bayesian statistics, posterior distributions obtained with Bayes' theorem often entail a high degree of complexity and are typically not analytically tractable. To circumvent this problem, MCMC methods have been developed for generating samples from (possibly unnormalized) target distributions [26]. One of the most popular MCMC methods together with the Gibbs sampler [27] is the Metropolis-Hastings (MH) algorithm originally proposed by [28] and later generalized by [29]. In this section, we propose to implement a modified version of the Metropolis-adjusted Langevin algorithm (MALA) [30] within the EpiLPS framework. The major advantage of MALA as compared to MH algorithms is that the proposal distribution is based upon a discretized approximation of the Langevin diffusion that uses the gradient of the target posterior distribution. These "smarter" proposals make use of additional information about the target density so that algorithms based on Langevin dynamics can converge at sub-geometric rates and tend to be more efficient than naive random-walk Metropolis algorithms [31, 32].

This motivates our choice for embedding a MALA algorithm in EpiLPS as an efficient way of obtaining MCMC samples for inference on the instantaneous reproduction number $\mathcal{R}_t$ via the renewal equation. The end-user will thus have a fully flexible choice regarding the underlying approach for estimating $\mathcal{R}_t$ either via Laplacian-P-splines, where the uncertainty surrounding the parameter $\lambda$ responsible for smoothing is ignored and $\lambda$ is fixed at its maximum *a posteriori* (LPSMAP); or via a modified MALA algorithm, where the uncertainty surrounding the penalty (and overdispersion) parameter is fully taken into account (LPSMALA). The approach permits to obtain samples from the joint posterior of the spline vector and the penalty and overdispersion

parameters. The latter can then be injected in functionals of the spline vector to obtain smooth estimates of the epidemic curve as well as the instantaneous reproduction number. Another advantage is that highest posterior density intervals can be easily calculated with LPSMALA.

**Conditional posteriors for a "Metropolis-within-Gibbs".   Joint posterior of $(\zeta, \lambda)$**

Let $\zeta = (\boldsymbol{\theta}^\top, \rho)^\top$ be the $(K + 1)$-dimensional vector gathering the B-spline coefficients $\boldsymbol{\theta}$ and the overdispersion parameter $\rho$. Using Bayes' theorem, the joint posterior distribution for $\zeta, \lambda$ and $\delta$ is:

$$
\begin{aligned}
p(\zeta, \lambda, \delta | \mathcal{D}) &= \frac{p(\mathcal{D}|\zeta, \lambda, \delta)p(\zeta, \lambda, \delta)}{p(\mathcal{D})} \\
&\propto \mathcal{L}(\zeta; \mathcal{D})p(\boldsymbol{\theta}|\lambda)p(\lambda|\delta)p(\delta)p(\rho) \\
&\propto \exp(\ell(\zeta; \mathcal{D}))p(\boldsymbol{\theta}|\lambda)p(\lambda|\delta)p(\delta)p(\rho).
\end{aligned}
\tag{15}
$$

The analytical formulas of the chosen priors are:

$$
\begin{aligned}
p(\boldsymbol{\theta}|\lambda) &\propto \lambda^{\frac{K}{2}} \exp(-0.5\lambda\boldsymbol{\theta}^\top P\boldsymbol{\theta}), \\
p(\lambda|\delta) &\propto \delta^{\frac{\phi}{2}}\lambda^{\frac{\phi}{2}-1} \exp(-0.5\phi\delta\lambda), \\
p(\delta) &\propto \delta^{a_\delta-1} \exp(-b_\delta\delta), \\
p(\rho) &\propto \rho^{a_\rho-1} \exp(-b_\rho\rho).
\end{aligned}
$$

Injecting the above priors into (15) yields:

$$
\begin{aligned}
p(\zeta, \lambda, \delta | \mathcal{D}) &\propto \exp\big(\ell(\zeta; \mathcal{D}) - b_\rho\rho - 0.5\lambda\boldsymbol{\theta}^\top P\boldsymbol{\theta}\big)\rho^{a_\rho-1}\lambda^{\frac{K+\phi}{2}-1} \\
&\times \delta^{\left(\frac{\phi}{2}+a_\delta\right)-1} \exp(-\delta(0.5\phi\lambda + b_\delta)).
\end{aligned}
\tag{16}
$$

**Conditional posteriors of $\zeta, \lambda$ and $\delta$**

The following conditional posterior distributions can be directly obtained from (16):

$$
p(\zeta|\lambda, \delta, \mathcal{D}) \propto \rho^{a_\rho-1} \exp(\ell(\zeta; \mathcal{D}) - b_\rho\rho - 0.5\lambda\boldsymbol{\theta}^\top P\boldsymbol{\theta}),
\tag{17}
$$

$$
(\lambda|\zeta, \delta, \mathcal{D}) \sim \mathcal{G}(0.5(K + \phi), 0.5(\boldsymbol{\theta}^\top P\boldsymbol{\theta} + \delta\phi)),
\tag{18}
$$

$$
(\delta|\zeta, \lambda, \mathcal{D}) \sim \mathcal{G}(0.5\phi + a_\delta, 0.5\phi\lambda + b_\delta).
\tag{19}
$$

**Sampling from the joint posterior $p(\zeta, \lambda, \delta | \mathcal{D})$**

As the full conditionals $p(\zeta|\lambda, \delta, \mathcal{D})$, $p(\lambda|\zeta, \delta, \mathcal{D})$ and $p(\delta|\zeta, \lambda, \mathcal{D})$ are available, we follow a "Metropolis-within-Gibbs" strategy to sample the joint posterior $p(\zeta, \lambda, \delta | \mathcal{D})$. In particular, the hyperparameters $\lambda$ and $\delta$ will be sampled in a Gibbs step, while $\zeta$ will be sampled using a modified Langevin-Hastings algorithm. This approach is presented in [33] in the context of Bayesian density estimation (see also [34] for the use of MALA in a proportional hazards model and [35] for a recent implementation in mixture cure models). We adapt the algorithm of the latter reference to our EpiLPS methodology. In particular, the variance-covariance matrix in the Langevin diffusion will be replaced by the variance-covariance matrix obtained with LPSMAP. The correlation structure borrowed from LPSMAP improves convergence and chain mixing.

**The modified Metropolis-adjusted Langevin algorithm.**   In what follows, we prefer to work under the $\log(\cdot)$ parameterization for $\rho$, i.e. $w = \log(\rho)$ and denote by $\widetilde{\zeta} = (\boldsymbol{\theta}^\top, w)^\top$, the $(K + 1)$-dimensional vector of B-spline amplitudes and (log) overdispersion $w$. Under this parameterization, the conditional posterior of $\widetilde{\zeta}$ given $\lambda$ and $\delta$ can be obtained from (17) by

using the transformation method of random variables:

$$p(\widetilde{\zeta}|\lambda, \delta, \mathcal{D}) \propto \exp(w)^{a_\rho} \exp(\ell(\widetilde{\zeta}; \mathcal{D}) - b_\rho \exp(w) - 0.5\lambda\boldsymbol{\theta}^\top P\boldsymbol{\theta}), \tag{20}$$

with the following log-likelihood under the reparameterization:

$$\ell(\widetilde{\zeta}; \mathcal{D}) \doteq \sum_{t=1}^{T} \left\{ \log\Gamma(y_t + \exp(w)) - \log\Gamma(\exp(w)) + y_t\boldsymbol{\theta}^\top b(t) + \exp(w)w \right.$$
$$\left. - (y_t + \exp(w))\log\left(\exp(\boldsymbol{\theta}^\top b(t)) + \exp(w)\right) \right\}. \tag{21}$$

Let us denote by $\widetilde{\zeta}^{(m-1)} \in \mathbb{R}^{(K+1)}$ the state of the chain at iteration $(m-1)$. In the Langevin-Hastings algorithm, the proposal for the vector $\widetilde{\zeta}$ at iteration $m$ is a draw from the following multivariate Gaussian distribution:

$$\widetilde{\zeta}^{(\text{prop})} \sim \mathcal{N}_{(K+1)}\left(\widetilde{\zeta}^{(m-1)} + 0.5\varrho\Sigma_{LH}\nabla_{\widetilde{\zeta}} \log p(\widetilde{\zeta}|\lambda, \delta, \mathcal{D})|_{\widetilde{\zeta} = \widetilde{\zeta}^{(m-1)}}, \varrho\Sigma_{LH}\right), \tag{22}$$

where $\varrho > 0$ is a tuning parameter that has to be carefully chosen in order to reach a desired acceptance rate and $\Sigma_{LH}$ is the following block-diagonal variance-covariance matrix:

$$\Sigma_{LH} = \begin{pmatrix} \Sigma^* & 0 \\ 0 & 1 \end{pmatrix}, \tag{23}$$

where $\Sigma^*$ is the $K$-dimensional covariance matrix obtained with LPSMAP. The gradient of $\log p(\widetilde{\zeta}|\lambda, \delta, \mathcal{D}) = \ell(\widetilde{\zeta}; \mathcal{D}) - 0.5\lambda\boldsymbol{\theta}^\top P\boldsymbol{\theta} - b_\rho \exp(w) + a_\rho w$ can be decomposed as follows:

$$\nabla_{\widetilde{\zeta}} \log p(\widetilde{\zeta}|\lambda, \delta, \mathcal{D}) = \left( \nabla_{\boldsymbol{\theta}}^\top \log p(\widetilde{\zeta}|\lambda, \delta, \mathcal{D}), \frac{\partial \log p(\widetilde{\zeta}|\lambda, \delta, \mathcal{D})}{\partial w} \right)^\top, \tag{24}$$

and is analytically available (see S1 Appendix for more details). All the quantities related to the Langevin-Hastings proposal have been analytically derived, so that the draw in (22) can be obtained (for a given value of $\lambda$ and $\delta$). As in a classic MH algorithm, the next step consists in computing the acceptance probability:

$$\pi\left(\widetilde{\zeta}^{(m-1)}, \widetilde{\zeta}^{(\text{prop})}\right) = \min\left\{ 1, \frac{p(\widetilde{\zeta}^{(\text{prop})}|\lambda, \delta, \mathcal{D})}{p(\widetilde{\zeta}^{(m-1)}|\lambda, \delta, \mathcal{D})} \frac{q\left(\widetilde{\zeta}^{(\text{prop})}, \widetilde{\zeta}^{(m-1)}\right)}{q\left(\widetilde{\zeta}^{(m-1)}, \widetilde{\zeta}^{(\text{prop})}\right)} \right\}, \tag{25}$$

where $q(\cdot, \cdot)$ denotes the (Gaussian) proposal distribution and $p(\cdot|\lambda, \delta, \mathcal{D})$ the target (conditional) posterior distribution. Finally, we generate a uniform random variable $u \sim \mathcal{U}(0, 1)$ and accept the proposed vector $\widetilde{\zeta}^{(\text{prop})}$ if $u \leq \pi(\widetilde{\zeta}^{(m-1)}, \widetilde{\zeta}^{(\text{prop})})$ and reject it otherwise. While iterating through the Metropolis-within-Gibbs algorithm, the tuning parameter $\varrho$ is automatically adapted to reach the optimal acceptance rate of 0.57 [31, 36, 37]. The pseudo-code below summarizes the LPSMALA algorithm.

**LPSMALA algorithm to sample the posterior $p(\boldsymbol{\theta}, \rho, \lambda, \delta|\mathcal{D})$.**

```
1: Fix initial values m = 0, λ⁽⁰⁾, δ⁽⁰⁾, ϱ⁽⁰⁾ and ζ̃⁽⁰⁾ = (θ⁽⁰⁾ᵀ, w⁽⁰⁾)ᵀ.
2:   for m = 1, ..., M do
3:   (Langevin-Hastings)
4:   Compute Langevin diffusion:
        ℰ(ζ̃⁽ᵐ⁻¹⁾) = ζ̃⁽ᵐ⁻¹⁾ + 0.5ϱ⁽ᵐ⁻¹⁾Σ_LH∇_ζ̃ log p(ζ̃|λ⁽ᵐ⁻¹⁾, δ⁽ᵐ⁻¹⁾, 𝒟)|_ζ̃=ζ̃⁽ᵐ⁻¹⁾ .
```

5:   Generate a proposal: $\widetilde{\boldsymbol{\zeta}}^{(\text{prop})} \sim \mathcal{N}_{(K+1)}\left(\mathscr{E}(\widetilde{\boldsymbol{\zeta}}^{(m-1)}), \varrho^{(m-1)}\boldsymbol{\Sigma}_{LH}\right).$

6:   Compute acceptance probability:

$$\pi\left(\widetilde{\boldsymbol{\zeta}}^{(m-1)}, \widetilde{\boldsymbol{\zeta}}^{(\text{prop})}\right) = \min\left\{1, \frac{p(\tilde{\zeta}^{(\text{prop})}|\lambda^{(m-1)},\delta^{(m-1)},\mathcal{D})}{p(\tilde{\zeta}^{(m-1)}|\lambda^{(m-1)},\delta^{(m-1)},\mathcal{D})}\frac{q\left(\tilde{\zeta}^{(\text{prop})},\tilde{\zeta}^{(m-1)}\right)}{q\left(\tilde{\zeta}^{(m-1)},\tilde{\zeta}^{(\text{prop})}\right)}\right\}.$$

7:   Draw $u \sim \mathcal{U}(0,1).$

8:   **if** $u \le \pi$ set $\widetilde{\boldsymbol{\zeta}}^{(m)} = \widetilde{\boldsymbol{\zeta}}^{(\text{prop})}$ (accept), **else** $\widetilde{\boldsymbol{\zeta}}^{(m)} = \widetilde{\boldsymbol{\zeta}}^{(m-1)}$ (reject).

9:   (*Gibbs sampler*)

10:    Draw $\delta^{(m)} \sim \mathcal{G}\left(0.5\phi + a_\delta, 0.5\phi\lambda^{(m-1)} + b_\delta\right),$

11:    Draw $\lambda^{(m)} \sim \mathcal{G}\left(0.5(K+\phi), 0.5(\boldsymbol{\theta}^{(m)\top}P\boldsymbol{\theta}^{(m)} + \delta^{(m)}\phi)\right).$

12:    (*Adaptive tuning*)

13:    Update $\varrho^{(m)} = \mathscr{H}^2\left(\sqrt{\varrho^{(m-1)}} + m^{-1}\left(\pi\left(\widetilde{\boldsymbol{\zeta}}^{(m-1)}, \widetilde{\boldsymbol{\zeta}}^{(\text{prop})}\right) - 0.57\right)\right).$

14: **end for**

The adaptive tuning part (line 13) involves the step function $\mathscr{H}(z) = \epsilon\mathbb{I}(z < \epsilon) + z\mathbb{I}(\epsilon \le z \le \mathscr{A}) + \mathscr{A}\mathbb{I}(z > \mathscr{A})$, with $\epsilon = 10^{-4}$ and $\mathscr{A} = 10^4$, see [33] for details. Finally, the ratio $q(\widetilde{\boldsymbol{\zeta}}^{(\text{prop})}, \widetilde{\boldsymbol{\zeta}}^{(m-1)})q^{-1}(\widetilde{\boldsymbol{\zeta}}^{(m-1)}, \widetilde{\boldsymbol{\zeta}}^{(\text{prop})})$ entering the computation of the acceptance probability (line 6) is derived in S1 Appendix.

**Posterior inference with LPSMALA.**   Provided the LPSMALA algorithm is iterated long enough, say after $\widetilde{M}$ iterations, MCMC theory certifies that $\mathscr{S} = \{(\boldsymbol{\theta}^{(m)\top}, \rho^{(m)}, \lambda^{(m)}, \delta^{(m)})\}_{m=\tilde{M}+1}^{M}$ can be viewed as random draws from the target posterior distribution $p(\boldsymbol{\theta}, \rho, \lambda, \delta|\mathcal{D})$. Note that a convenient starting point for the initial values of the parameters might be to fix them at their LPSMAP estimate. Given the sample $\mathscr{S}$, inference on quantities that are functions of $\boldsymbol{\theta}$ becomes straightforward in the sense that point estimates and credible intervals can be easily obtained. A point estimate for the mean number of incidence counts at time $t$ is taken to be the posterior mean (after discarding the burn-in phase):

$$\hat{\mu}(t) = \frac{1}{M - \widetilde{M}} \sum_{m=\tilde{M}+1}^{M} \exp\left(\boldsymbol{\theta}^{(m)\top}b(t)\right). \tag{26}$$

Note also that $\mathscr{S}$ can be used to compute highest posterior density intervals of $\mu(t)$ at any point $t$. Using the renewal equation and the MCMC sample, one can apply the "plug-in" method and recover the following estimate of the instantaneous reproduction number at time point $t$:

$$\hat{\mathcal{R}}_t = \frac{1}{M - \widetilde{M}} \sum_{m=\tilde{M}+1}^{M} \exp\left(\boldsymbol{\theta}^{(m)\top}b(t)\right)\left\{\mathbb{I}(t=1) + \left(\sum_{s=1}^{t-1}\varphi_s \exp\left(\boldsymbol{\theta}^{(m)\top}b(t-s)\right)\right)^{-1}\mathbb{I}(2 \le t \le k)\right.$$
$$\left. + \left(\sum_{s=1}^{k}\varphi_s \exp\left(\boldsymbol{\theta}^{(m)\top}b(t-s)\right)\right)^{-1}\mathbb{I}(k < t \le T)\right\}. \tag{27}$$

Also, using $\mathscr{S}$, one can compute a highest posterior density interval of $\mathcal{R}_t$ at time step $t$.

## Results

### Setting of the simulation study

In this section, a numerical study is implemented with nine epidemic scenarios to assess the accuracy with which EpiLPS is able to track the target reproduction number over time. EpiLPS results are compared with the instantaneous reproduction number estimate from the EpiEstim package [3] using three sliding windows options (the default weekly windows, three days windows and daily windows). In addition, we disentangle between comparisons of EpiLPS against

**Table 1. Time domain of the epidemic curve, assumed functional form of the reproduction number, serial interval distribution and its associated source(s) in the literature for the nine scenarios considered in the simulation study.**

| Scenario | Time domain of epidemic curve | $\mathcal{R}_t$ target function | Serial Interval Mean (SD), days | Reference for Serial Interval |
|---|---|---|---|---|
| 1 | $\mathcal{T} = [1, 40]$ | $\mathcal{R}_t = 1.3$ | $\varphi_{FLU}$ 2.6 (1.5) | Ferguson et al. (2005) [38] Cori et al. (2013) [3] |
| 2 | | $\mathcal{R}_t = 2 \ \mathbb{I}(t < 20) + 0.9 \ \mathbb{I}(t \geq 20)$ | | |
| 3 | | $\mathcal{R}_t = 0.25 + \exp(\cos(t/7))$ | | |
| 4 | | $\mathcal{R}_t = \exp(\cos(t/15))$ | | |
| 5 | $\mathcal{T} = [1, 40]$ | $\mathcal{R}_t = 1.3$ | $\varphi_{SARS}$ 8.4 (3.8) | Lipsitch et al. (2003) [39] Cori et al. (2013) [3] |
| 6 | | $\mathcal{R}_t = 2 \ \mathbb{I}(t < 20) + 0.9 \ \mathbb{I}(t \geq 20)$ | | |
| 7 | | $\mathcal{R}_t = 0.25 + \exp(\cos(t/7))$ | | |
| 8 | | $\mathcal{R}_t = \exp(\cos(t/15))$ | | |
| 9 | $\mathcal{T} = [1, 60]$ | $\mathcal{R}_t = 0.5 \ (\exp(\sin(\pi t/9)) + 1.5 \exp(\cos(4/t)))$ | $\varphi_{MERS}$ 6.8(4.1) | Cauchemez et al. (2016) [40] |

EpiEstim with $\mathcal{R}_t$ estimates reported on the last day of a window following the convention of [3] and $\mathcal{R}_t$ estimates reported at the midpoint of a smoothing window following the best practice recommendation of [2]. For EpiLPS, $K = 40$ (cubic) B-splines are specified with a second-order penalty and a chain length of 3 000 for LPSMALA (including a burn-in of size 1 000). In each scenario, $S = 100$ incidence time series of $T$ days are simulated (initiated with 10 index cases). The epidemic data generating process computes the mean incidence at a given day $t$, i.e. $\mu(t)$ according to the renewal equation and the incidence of cases at time point $t$ is sampled from the negative binomial distribution $y_t \sim \text{NegBin}(\mu(t), \rho)$. The simulation study also accounts for varying degrees of overdispersion by using different values of $\rho$ in the considered scenarios. Furthermore, the incidence data are generated according to three different serial interval distributions, namely $\varphi_{FLU}$, $\varphi_{SARS}$ and $\varphi_{MERS}$ corresponding to an influenza, a SARS-CoV-1 and a MERS-CoV like serial interval, respectively. The discretized version of the SI distributions are computed by using the Cori et al. (2013) [3] discretization formula assuming a (shifted) Gamma distribution. In Scenario 1, a constant instantaneous reproduction number $\mathcal{R}_t = 1.3$ is considered. Scenario 2 imitates an intervention strategy, so that $\mathcal{R}_t = 2$ until a sudden drop to $\mathcal{R}_t = 0.9$ occurs at day $t = 20$. The latter scenario allows to check whether EpiLPS is able to quickly react to a sudden change in $\mathcal{R}_t$. Scenario 3 is characterized by a more wiggly structure for $\mathcal{R}_t$ and Scenario 4 considers the case of a vanishing epidemic with a monotonic decreasing reproduction number. Scenarios 5–8 assume the same functional form for $\mathcal{R}_t$ as in Scenarios 1–4 but with a different serial interval distribution. In Scenario 9, the $\mathcal{R}_t$ function is chosen in such a way that there is a single large wave in the early phase of the epidemic and a more stable pattern (with smaller waves) in the late phase. Table 1 summarizes the time domain of the epidemic curve, the target $\mathcal{R}_t$ function, the serial interval distribution and its associated source(s) in the literature.

The simulation study is organized as follows. First, we compare EpiLPS with EpiEstim using the convention of Cori and colleagues, namely reporting the $\mathcal{R}_t$ estimate at the end of the smoothing window, which is well suited for real-time estimation. The latter approach reports the $\mathcal{R}_t$ estimate computed in the window $[t - \omega; t]$, where $\omega$ denotes the window width. Next, the Gostic et al. (2020) [2] recommendation is used, where the $\mathcal{R}_t$ estimate is reported at the center of the window, i.e. $[t - \omega/2; t + \omega/2]$. Concentrating on the window midpoint avoids lagged $\mathcal{R}_t$ estimates at the cost of ruling out estimates at the last $\omega/2$ time points as, in that case, the upper bound of the window reaches future calendar days for which data is not yet available. Fig 1 summarizes the two window structures used in the simulation study.

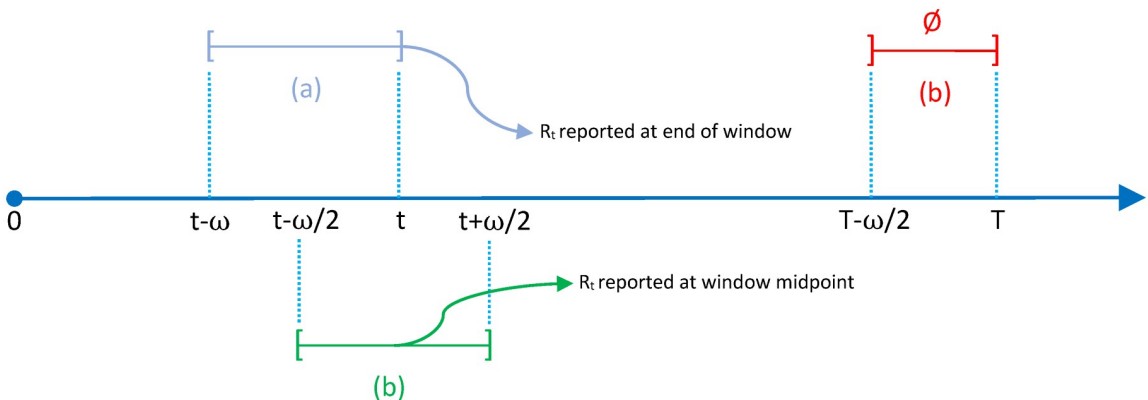

**Fig 1.** Illustration of smoothing windows of width $\omega$ to estimate $\mathcal{R}_t$ with EpiEstim. (a) Cori et al. (2013) [3] convention with sliding windows $[t - \omega; t]$, where $\mathcal{R}_t$ is reported at the end of the window. (b) Gostic et al. (2020) [2] recommendation with centered sliding windows $[t - \omega/2; t + \omega/2]$, where $\mathcal{R}_t$ is reported at the midpoint of the window. Under the midpoint rule, $\mathcal{R}_t$ estimates for the last $\omega/2$ time units are unavailable $\emptyset$.

## Comparing EpiLPS with EpiEstim at window boundary

The performance indicators computed for each scenario include the average bias, mean square error (MSE), coverage probability (*CP*) and width (*CI*$^\Delta$) of 90% and 95% credible intervals for the $\mathcal{R}_t$ estimator (see detailed formulas in S2 Appendix) with EpiLPS and EpiEstim, respectively. Estimates obtained during the first week of the epidemic are ignored as they may be subject to serious bias due to the poor information carried by the (few) incident cases in such an early phase. Therefore, the performance measures are computed as an average over days $t = 8, \ldots, T$, where $T$ is the upper bound of $\mathcal{T}$. For a chosen time window, the performance measures for EpiEstim are computed by comparing the true value of the reproduction number at time step $t$ with the estimated reproduction number (and credible interval) obtained at the end of the chosen time window (cf. Fig 1). A detailed description of the data generating process and the figures of the estimated $\mathcal{R}_t$ trajectories for all scenarios are provided in S2 Appendix.

The performance measures given in Tables 2 and 3 provide interesting insights into the behavior of EpiLPS and EpiEstim across the considered scenarios. In terms of bias, EpiLPS is really competitive against EpiEstim as both LPSMAP and LPSMALA outperform EpiEstim (no matter the time window size) in Scenarios 4–8. For the remaining scenarios, the bias between the two competing methods is more or less similar. Regarding the MSE, EpiLPS exhibits smaller values as compared to EpiEstim with three days and daily windows respectively across all scenarios. Moreover, specifying smaller time windows in EpiEstim leads (generally) to an increase in MSE and also an increase in bias. A close inspection of the coverage probability of credible intervals reveals that EpiLPS has close to nominal coverage in almost all scenarios. This is however not the case for EpiEstim, especially for weekly and three days windows, where severe to mild undercoverage is observed. Also, EpiEstim tends to show more severe undercoverage in scenarios where data is more overdispersed (see e.g. Scenario 4). More importantly, even when EpiEstim approaches the nominal coverage probability (with daily windows), it has much wider credible interval width (and so less precision) as compared to EpiLPS in almost all scenarios.

Figs 2 to 4 summarize the epidemic curves and the trajectories obtained for the estimated $\mathcal{R}_t$ with LPSMAP (blue curves) and EpiEstim under weekly sliding windows (green curves) for selected scenarios. These figures highlight the flexibility and the precision with which Laplacian-P-splines are able to capture the reproduction number over the time course of the

**Table 2. Results for EpiLPS and EpiEstim in Scenarios 1–8 for $S$ = 100 simulated epidemics.** The Bias, MSE, coverage probability ($CP$) and width ($CI^\Delta$) of 90% and 95% credible intervals for $\mathcal{R}_t$ are averaged over days $t = 8, \ldots, 40$. For EpiEstim, $\mathcal{R}_t$ is reported at the end of the window.

| Scenario | Method | Bias | MSE | $CP_{90\%}$ | $CP_{95\%}$ | $CI^\Delta_{90\%}$ | $CI^\Delta_{95\%}$ |
|---|---|---|---|---|---|---|---|
| 1 $\varphi_{FLU}$ | LPSMAP | -0.012 | 0.016 | 90.970 | 95.333 | 0.399 | 0.477 |
| | LPSMALA | -0.012 | 0.017 | 93.000 | 96.727 | 0.454 | 0.544 |
| | EpiEstim (7d windows) | -0.007 | 0.012 | 88.970 | 94.242 | 0.330 | 0.394 |
| | EpiEstim (3d windows) | 0.004 | 0.025 | 88.636 | 93.515 | 0.466 | 0.555 |
| | EpiEstim (1d windows) | 0.036 | 0.075 | 88.515 | 93.970 | 0.776 | 0.927 |
| 2 $\varphi_{FLU}$ | LPSMAP | -0.004 | 0.013 | 77.030 | 82.424 | 0.175 | 0.209 |
| | LPSMALA | -0.003 | 0.010 | 92.697 | 93.909 | 0.313 | 0.375 |
| | EpiEstim (7d windows) | 0.071 | 0.043 | 67.970 | 73.303 | 0.142 | 0.169 |
| | EpiEstim (3d windows) | 0.027 | 0.021 | 77.818 | 84.424 | 0.182 | 0.217 |
| | EpiEstim (1d windows) | 0.003 | 0.016 | 83.424 | 89.242 | 0.287 | 0.342 |
| 3 $\varphi_{FLU}$ | LPSMAP | -0.009 | 0.015 | 91.970 | 96.242 | 0.396 | 0.474 |
| | LPSMALA | -0.008 | 0.017 | 92.545 | 96.152 | 0.451 | 0.541 |
| | EpiEstim (7d windows) | -0.023 | 0.093 | 23.394 | 29.333 | 0.279 | 0.332 |
| | EpiEstim (3d windows) | -0.006 | 0.033 | 76.939 | 84.939 | 0.434 | 0.518 |
| | EpiEstim (1d windows) | 0.044 | 0.073 | 89.576 | 94.394 | 0.761 | 0.909 |
| 4 $\varphi_{FLU}$ | LPSMAP | 0.000 | 0.002 | 89.879 | 94.091 | 0.189 | 0.226 |
| | LPSMALA | -0.001 | 0.003 | 92.152 | 96.455 | 0.175 | 0.208 |
| | EpiEstim (7d windows) | 0.152 | 0.027 | 9.485 | 11.273 | 0.105 | 0.125 |
| | EpiEstim (3d windows) | 0.056 | 0.007 | 40.970 | 47.667 | 0.133 | 0.158 |
| | EpiEstim (1d windows) | 0.003 | 0.008 | 80.848 | 88.455 | 0.209 | 0.250 |
| 5 $\varphi_{SARS}$ | LPSMAP | 0.013 | 0.232 | 93.030 | 96.667 | 1.903 | 2.356 |
| | LPSMALA | 0.005 | 0.246 | 91.061 | 96.273 | 1.897 | 2.376 |
| | EpiEstim (7d windows) | 0.061 | 0.162 | 83.344 | 89.781 | 1.089 | 1.301 |
| | EpiEstim (3d windows) | 0.146 | 0.395 | 82.844 | 89.438 | 1.619 | 1.939 |
| | EpiEstim (1d windows) | 0.388 | 1.091 | 86.938 | 92.281 | 2.783 | 3.362 |
| 6 $\varphi_{SARS}$ | LPSMAP | 0.001 | 0.201 | 90.939 | 95.394 | 1.660 | 2.044 |
| | LPSMALA | -0.012 | 0.218 | 91.485 | 96.242 | 1.731 | 2.157 |
| | EpiEstim (7d windows) | 0.115 | 0.202 | 72.562 | 79.812 | 0.909 | 1.085 |
| | EpiEstim (3d windows) | 0.114 | 0.330 | 77.719 | 85.219 | 1.295 | 1.548 |
| | EpiEstim (1d windows) | 0.238 | 0.843 | 81.312 | 88.219 | 2.162 | 2.602 |
| 7 $\varphi_{SARS}$ | LPSMAP | -0.008 | 0.288 | 96.121 | 98.606 | 2.272 | 3.023 |
| | LPSMALA | -0.005 | 0.384 | 92.182 | 96.545 | 1.929 | 2.485 |
| | EpiEstim (7d windows) | 0.040 | 0.278 | 75.062 | 82.781 | 1.090 | 1.304 |
| | EpiEstim (3d windows) | 0.160 | 0.410 | 85.125 | 91.656 | 1.700 | 2.044 |
| | EpiEstim (1d windows) | 0.493 | 1.199 | 89.562 | 94.031 | 3.051 | 3.707 |
| 8 $\varphi_{SARS}$ | LPSMAP | 0.021 | 0.187 | 91.667 | 95.303 | 1.416 | 1.750 |
| | LPSMALA | 0.005 | 0.191 | 91.061 | 96.121 | 1.522 | 1.900 |
| | EpiEstim (7d windows) | 0.217 | 0.206 | 69.312 | 77.281 | 0.838 | 1.000 |
| | EpiEstim (3d windows) | 0.157 | 0.311 | 79.375 | 86.781 | 1.163 | 1.391 |
| | EpiEstim (1d windows) | 0.250 | 0.725 | 82.594 | 89.281 | 1.950 | 2.350 |

epidemic. The dashed (dotted) curves represent the pointwise median (computed over the $S$ = 100 estimates) of $\mathcal{R}_t$ with LPSMAP (EpiEstim). For LPSMAP, it closely follows the true pattern of $\mathcal{R}_t$ even under strong nonlinearities as in Fig 4. The EpiEstim trajectories appear shifted to the right of the target $\mathcal{R}_t$ curve. This lag is due to the fact that for weekly sliding windows, the $\mathcal{R}_t$ estimate provided by EpiEstim at the end of the window is entirely based on data

**Table 3. Results for EpiLPS and EpiEstim in Scenario 9 for $S$ = 100 simulated epidemics.** The Bias, MSE, coverage probability ($CP$) and width ($CI^\Delta$) of 90% and 95% credible intervals for $\mathcal{R}_t$ are averaged over days $t = 8, \ldots, 60$. For EpiEstim, $\mathcal{R}_t$ is reported at the end of the window.

| Scenario | Method | Bias | MSE | $CP_{90\%}$ | $CP_{95\%}$ | $CI^\Delta_{90\%}$ | $CI^\Delta_{95\%}$ |
|---|---|---|---|---|---|---|---|
| 9 | LPSMAP | -0.026 | 0.078 | 90.566 | 94.528 | 0.849 | 1.021 |
| $\varphi_{MERS}$ | LPSMALA | -0.024 | 0.082 | 92.981 | 96.925 | 0.935 | 1.128 |
| | EpiEstim (7d windows) | -0.002 | 0.347 | 41.981 | 49.698 | 0.705 | 0.841 |
| | EpiEstim (3d windows) | 0.048 | 0.184 | 79.925 | 86.830 | 1.035 | 1.236 |
| | EpiEstim (1d windows) | 0.160 | 0.423 | 85.585 | 91.358 | 1.766 | 2.121 |

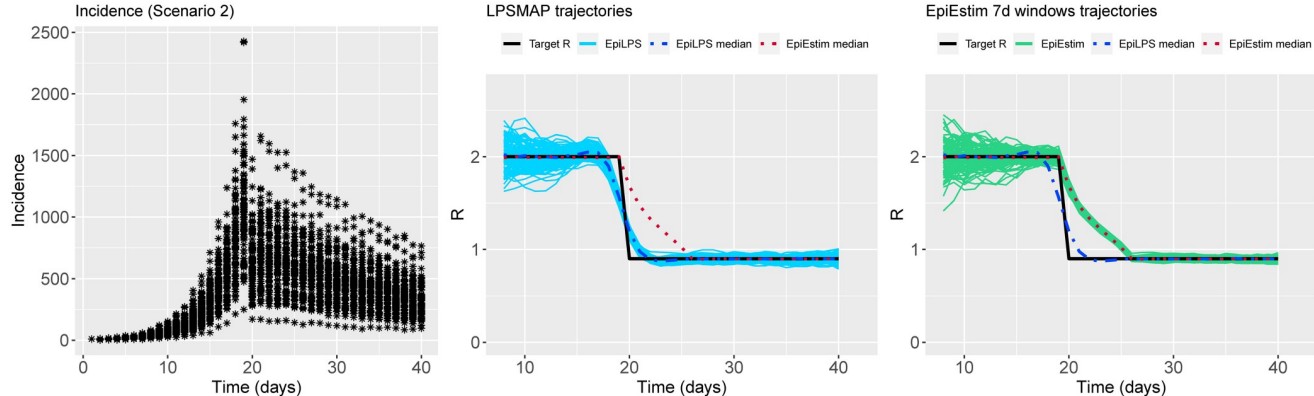

**Fig 2.** (Left) Simulated incidence data for Scenario 2. (Center) Estimated trajectories of $\mathcal{R}_t$ for each simulated dataset with LPSMAP. (Right) Estimated trajectories of $\mathcal{R}_t$ with EpiEstim using weekly sliding windows and $\mathcal{R}_t$ reported at the end of the window. The pointwise median estimate of $\mathcal{R}_t$ for EpiLPS (dashed) and EpiEstim (dotted) is also shown.

from past days and is therefore lagged compared to the target (instantaneous) $\mathcal{R}_t$. This shift effect can be corrected by decreasing the time window (e.g., using daily windows) at the cost of more "noisy" trajectories. Even then, the median $\mathcal{R}_t$ estimates of EpiEstim appear to capture less precisely the target $\mathcal{R}_t$ function as compared to LPSMAP/LPSMALA (see S2 Appendix) across most of the considered scenarios.

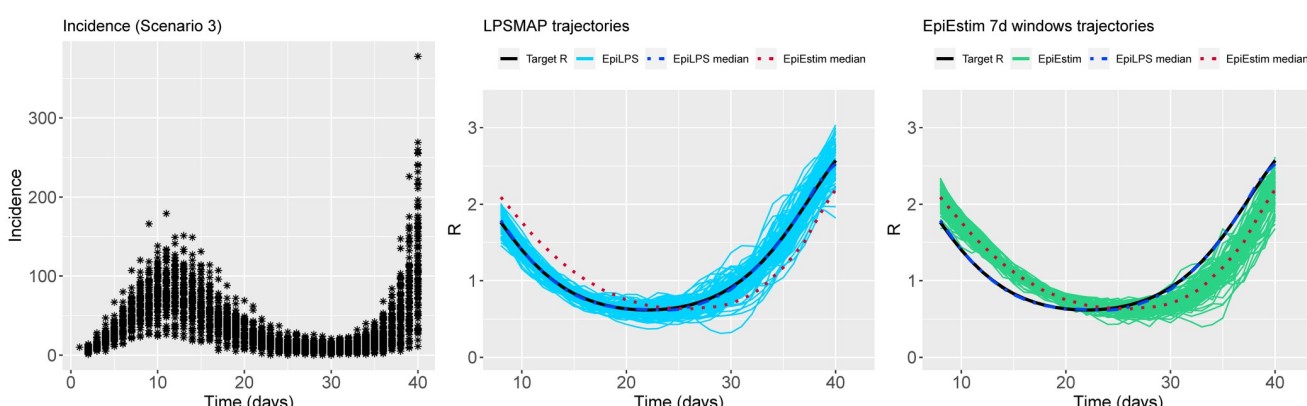

**Fig 3.** (Left) Simulated incidence data for Scenario 3. (Center) Estimated trajectories of $\mathcal{R}_t$ for each simulated dataset with LPSMAP. (Right) Estimated trajectories of $\mathcal{R}_t$ with EpiEstim using weekly sliding windows and $\mathcal{R}_t$ reported at the end of the window. The pointwise median estimate of $\mathcal{R}_t$ for EpiLPS (dashed) and EpiEstim (dotted) is also shown.

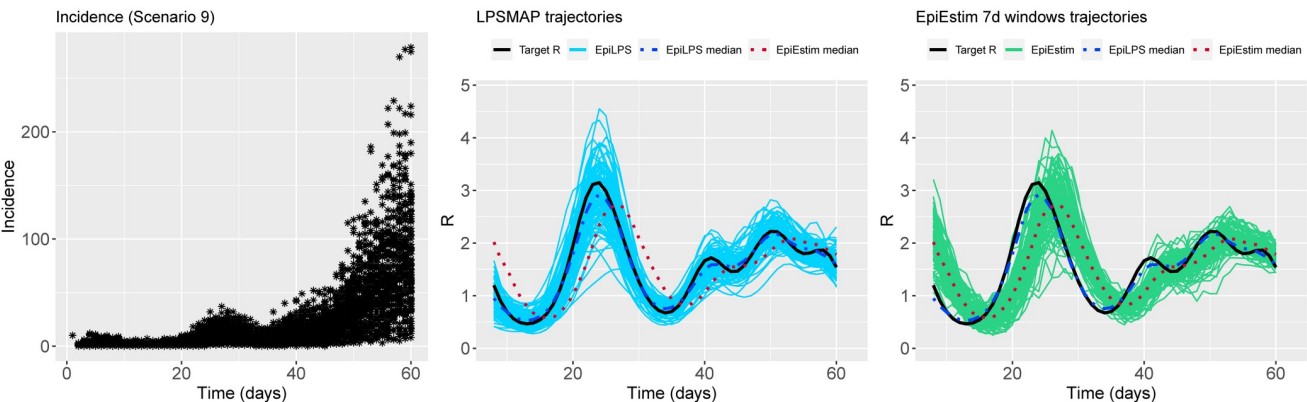

**Fig 4.** (Left) Simulated incidence data for Scenario 9. (Center) Estimated trajectories of $\mathcal{R}_t$ for each simulated dataset with LPSMAP. (Right) Estimated trajectories of $\mathcal{R}_t$ with EpiEstim using weekly sliding windows and $\mathcal{R}_t$ reported at the end of the window. The pointwise median estimate of $\mathcal{R}_t$ for EpiLPS (dashed) and EpiEstim (dotted) is also shown.

To summarize, this simulation study sheds light on the trade-off faced by the Cori method when estimating the instantaneous reproduction number. Choosing a weekly sliding window as a default option in EpiEstim can lead to a forward shifted (and so inaccurate) estimate of $\mathcal{R}_t$. Smaller time windows in EpiEstim alleviate the lag effect, but the price to pay is that the fitted $\mathcal{R}_t$ trajectory is wiggly (undersmoothing) as it captures more variation than necessary [2].

EpiLPS does not suffer from such a trade-off as the latter is naturally solved by P-splines. In fact, one could say that the time window size in EpiEstim is analogue to the smoothing parameter λ in EpiLPS as these quantities will be key for the resulting smoothness of the fit. The major advantage with EpiLPS is that λ is estimated naturally within the Bayesian model (either via maximum *a posteriori* estimation or MCMC), while the choice of the time window in EpiEstim is chosen freely outside the model.

## Comparing EpiLPS with EpiEstim at window midpoint

To correct for the lag effect in EpiEstim resulting from reporting the reproduction number estimate at the end of the window, Gostic and colleagues recommend to report it at the center of the window to obtain an estimate that is more accurately oriented in time. It is therefore important to compare the performance of EpiLPS against this "corrected" EpiEstim output as it is considered a best practice for a retrospective usage and, as such, is a legitimate candidate against EpiLPS which is by nature only partially real-time (see next section). We therefore run the entire simulations for all scenarios one more time accounting for the corrected EpiEstim output under weekly windows $\omega = 6$ and three days windows $\omega = 2$, where the estimated $\mathcal{R}_t$ is computed at the window midpoint. Results for a daily window ($\omega = 0$) are identical to those reported in Tables 2 and 3, as sliding windows become degenerate intervals at each time step $[t, t]$. The performance measures are reported in Table 4 and Figs 5 to 7 summarize the estimated trajectories for the same scenarios as in the previous section for the sake of comparison. As expected, the resulting $\mathcal{R}_t$ trajectories for EpiEstim are now closer to the target and the lag effect has disappeared. Despite this improvement, the performance indicators clearly highlight that EpiLPS outperforms EpiEstim in all scenarios except Scenario 1, where the numbers are of a similar order of magnitude. In general, the EpiLPS approach is less biased and provides credible intervals with close to nominal coverage. Even when correcting for the reporting of $\mathcal{R}_t$ at the middle of the window, EpiEstim results are less accurate, especially regarding

**Table 4. Simulation results for EpiLPS and EpiEstim in Scenarios 1–9 for $S$ = 100 simulated epidemics.** The performance indicators in Scenarios 1–8 for $\mathcal{R}_t$ are averaged over days t = 8,...,37 for LPSMAP, LPSMALA and weekly windows (EpiEstim) and over days t = 8,...,39 for 3 days windows under EpiEstim. In Scenario 9, the performance indicators for $\mathcal{R}_t$ are averaged over days t = 8,...,57 for LPSMAP, LPSMALA and weekly windows (EpiEstim) and over days t = 8,...,59 for 3 days windows under EpiEstim. For EpiEstim, $\mathcal{R}_t$ is reported at the window midpoint.

| Scenario | Method | Bias | MSE | $CP_{90\%}$ | $CP_{95\%}$ | $CI^{\Delta}_{90\%}$ | $CI^{\Delta}_{95\%}$ |
|---|---|---|---|---|---|---|---|
| 1 $\varphi_{FLU}$ | LPSMAP | -0.013 | 0.017 | 91.433 | 95.567 | 0.415 | 0.497 |
| | LPSMALA | -0.013 | 0.018 | 93.033 | 96.567 | 0.467 | 0.560 |
| | EpiEstim (7d windows) | -0.010 | 0.011 | 88.533 | 93.900 | 0.303 | 0.361 |
| | EpiEstim (3d windows) | 0.001 | 0.023 | 88.531 | 93.406 | 0.452 | 0.539 |
| 2 $\varphi_{FLU}$ | LPSMAP | -0.004 | 0.015 | 75.900 | 81.267 | 0.179 | 0.214 |
| | LPSMALA | -0.003 | 0.011 | 92.300 | 93.467 | 0.322 | 0.386 |
| | EpiEstim (7d windows) | -0.032 | 0.036 | 65.733 | 71.167 | 0.106 | 0.126 |
| | EpiEstim (3d windows) | -0.007 | 0.015 | 77.500 | 84.156 | 0.166 | 0.198 |
| 3 $\varphi_{FLU}$ | LPSMAP | -0.009 | 0.013 | 92.033 | 96.267 | 0.368 | 0.440 |
| | LPSMALA | -0.010 | 0.014 | 92.233 | 95.967 | 0.407 | 0.487 |
| | EpiEstim (7d windows) | 0.020 | 0.016 | 79.233 | 86.500 | 0.268 | 0.319 |
| | EpiEstim (3d windows) | 0.014 | 0.023 | 89.031 | 94.094 | 0.431 | 0.514 |
| 4 $\varphi_{FLU}$ | LPSMAP | 0.000 | 0.002 | 89.200 | 93.667 | 0.192 | 0.230 |
| | LPSMALA | 0.000 | 0.003 | 92.300 | 96.367 | 0.179 | 0.214 |
| | EpiEstim (7d windows) | -0.029 | 0.005 | 34.800 | 43.200 | 0.074 | 0.089 |
| | EpiEstim (3d windows) | -0.005 | 0.003 | 80.781 | 87.281 | 0.119 | 0.142 |
| 5 $\varphi_{SARS}$ | LPSMAP | 0.016 | 0.227 | 93.567 | 96.933 | 1.897 | 2.345 |
| | LPSMALA | 0.001 | 0.234 | 91.400 | 96.433 | 1.887 | 2.359 |
| | EpiEstim (7d windows) | 0.057 | 0.160 | 82.967 | 89.333 | 1.065 | 1.272 |
| | EpiEstim (3d windows) | 0.146 | 0.395 | 82.844 | 89.438 | 1.619 | 1.939 |
| 6 $\varphi_{SARS}$ | LPSMAP | 0.003 | 0.203 | 91.233 | 95.433 | 1.682 | 2.067 |
| | LPSMALA | -0.016 | 0.217 | 91.967 | 96.467 | 1.764 | 2.193 |
| | EpiEstim (7d windows) | 0.010 | 0.154 | 74.967 | 82.400 | 0.856 | 1.022 |
| | EpiEstim (3d windows) | 0.080 | 0.316 | 78.469 | 86.000 | 1.295 | 1.548 |
| 7 $\varphi_{SARS}$ | LPSMAP | -0.003 | 0.187 | 96.267 | 98.733 | 1.895 | 2.401 |
| | LPSMALA | -0.024 | 0.193 | 92.467 | 96.667 | 1.649 | 2.096 |
| | EpiEstim (7d windows) | 0.082 | 0.155 | 84.433 | 91.267 | 1.049 | 1.256 |
| | EpiEstim (3d windows) | 0.185 | 0.410 | 86.219 | 92.125 | 1.700 | 2.044 |
| 8 $\varphi_{SARS}$ | LPSMAP | 0.023 | 0.198 | 91.700 | 95.267 | 1.447 | 1.773 |
| | LPSMALA | 0.004 | 0.201 | 92.067 | 96.733 | 1.595 | 1.987 |
| | EpiEstim (7d windows) | 0.026 | 0.125 | 78.067 | 86.733 | 0.770 | 0.919 |
| | EpiEstim (3d windows) | 0.095 | 0.294 | 80.906 | 87.531 | 1.163 | 1.391 |
| 9 $\varphi_{SARS}$ | LPSMAP | -0.026 | 0.081 | 91.340 | 95.240 | 0.870 | 1.047 |
| | LPSMALA | -0.025 | 0.084 | 93.200 | 97.140 | 0.948 | 1.142 |
| | EpiEstim (7d windows) | -0.004 | 0.087 | 77.180 | 84.920 | 0.675 | 0.805 |
| | EpiEstim (3d windows) | 0.047 | 0.141 | 85.788 | 91.500 | 1.024 | 1.223 |

credible intervals with weekly windows that can strongly undercover. This has important implications regarding the recommendation of using EpiLPS in practice and detailed recommendation guidelines are provided below.

## Real-time considerations

EpiEstim is a powerful tool to estimate $\mathcal{R}_t$ in real-time and is probably the best tool currently available to deliver timely estimates of the reproduction number [41]. EpiLPS can be

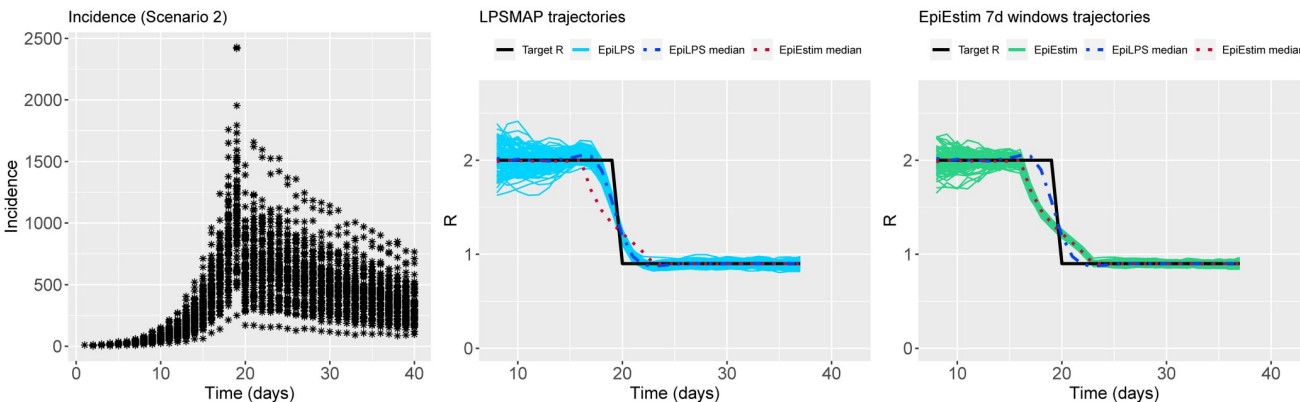

**Fig 5.** (Left) Simulated incidence data for Scenario 2. (Center) Estimated trajectories of $\mathcal{R}_t$ for each simulated dataset with LPSMAP. (Right) Estimated trajectories of $\mathcal{R}_t$ with EpiEstim using weekly sliding windows and $\mathcal{R}_t$ reported at the window midpoint. The pointwise median estimate of $\mathcal{R}_t$ for EpiLPS (dashed) and EpiEstim (dotted) is also shown.

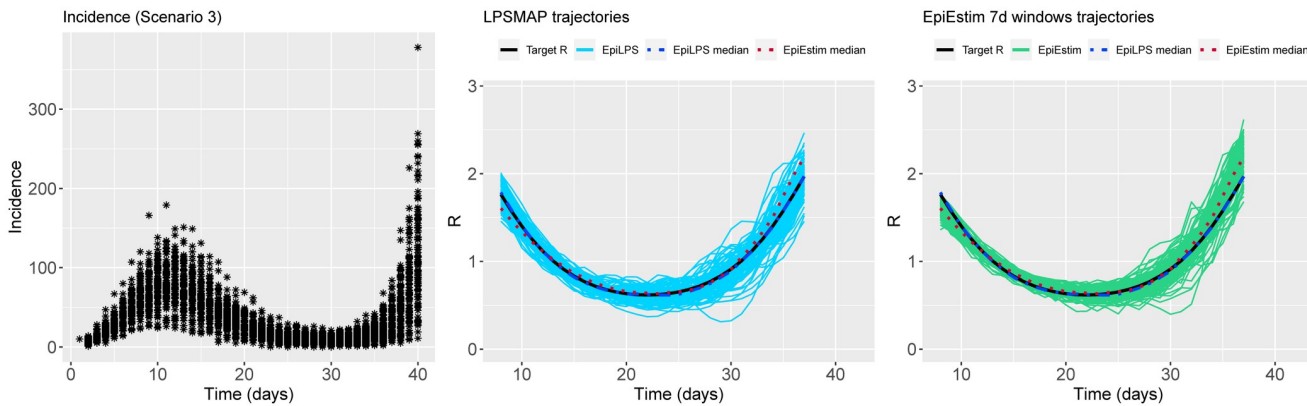

**Fig 6.** (Left) Simulated incidence data for Scenario 3. (Center) Estimated trajectories of $\mathcal{R}_t$ for each simulated dataset with LPSMAP. (Right) Estimated trajectories of $\mathcal{R}_t$ with EpiEstim using weekly sliding windows and $\mathcal{R}_t$ reported at the window midpoint. The pointwise median estimate of $\mathcal{R}_t$ for EpiLPS (dashed) and EpiEstim (dotted) is also shown.

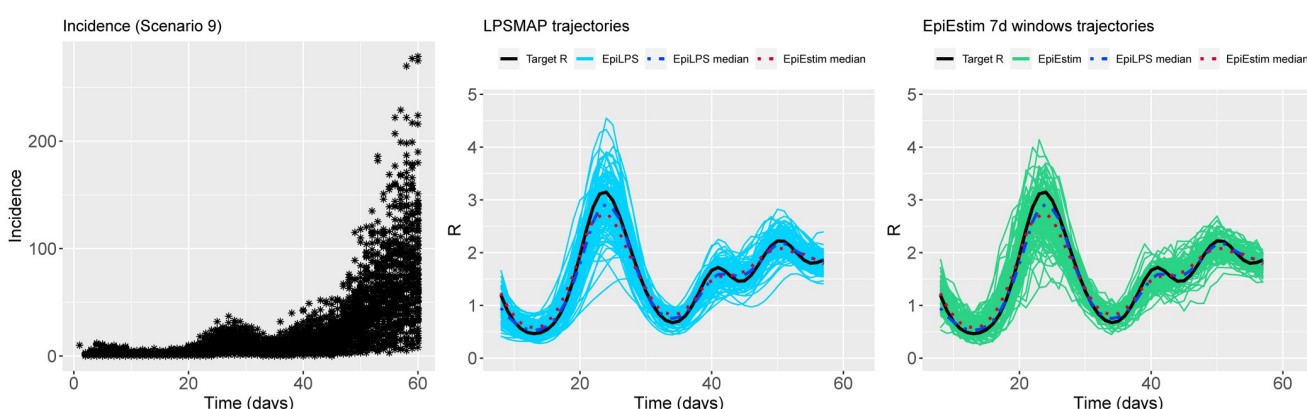

**Fig 7.** (Left) Simulated incidence data for Scenario 9. (Center) Estimated trajectories of $\mathcal{R}_t$ for each simulated dataset with LPSMAP. (Right) Estimated trajectories of $\mathcal{R}_t$ with EpiEstim using weekly sliding windows and $\mathcal{R}_t$ reported at the window midpoint. The pointwise median estimate of $\mathcal{R}_t$ for EpiLPS (dashed) and EpiEstim (dotted) is also shown.

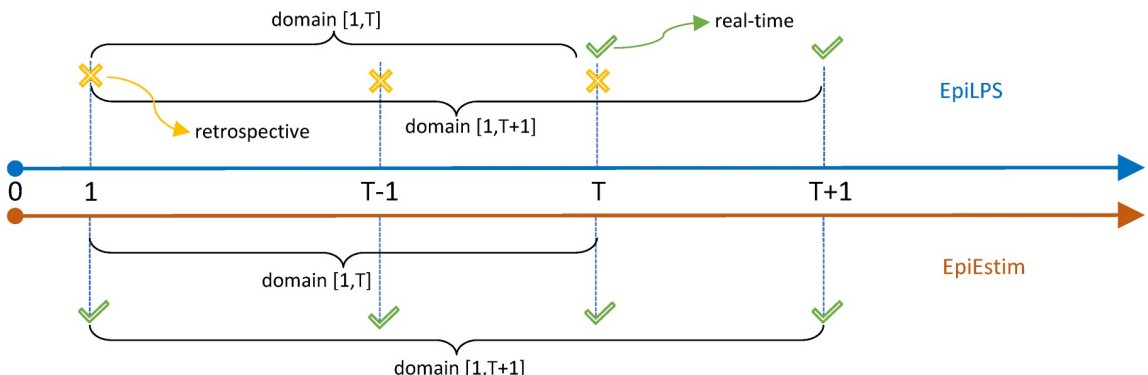

**Fig 8. Real-time properties of EpiLPS (top) and EpiEstim (bottom) when applied on domains $\mathcal{T} = [1, T]$ and $\mathcal{T}^* = [1, T + 1]$.**
EpiLPS provides real-time estimates of $\mathcal{R}_t$ only at the boundary of the considered domain and estimates at preceding time points are
retrospective. On the contrary, estimates of $\mathcal{R}_t$ with EpiEstim are always real-time and therefore preferred for a timely usage.

considered a real-time approach only to a certain degree, where the real-time concept is partially present but fundamentally different from the one proposed in EpiEstim. By real-time method, we mean a method for which an estimate of the reproduction number at time $t$ uses data up to (and including) time $t$. Let us assume that EpiLPS is applied on epidemic data over a specific period, say, $\mathcal{T} = [1, T]$. For time points $t = 1, \ldots, T − 1$, EpiLPS is clearly non real-time as the global smoothing of $\mathcal{R}_t$ on the "bandwidth" $\mathcal{T}$ will be computed based on past, current and future data values. However, at the domain boundary (time point $T$), the EpiLPS estimate of $\mathcal{R}_t$ will exclusively make use of data up to time $T$ and is therefore real-time (in the same sense as EpiEstim).

The EpiLPS real-time characteristic for this last time point is however only retained temporarily, as if applied (the next day) over the period $\mathcal{T}^* = [1, T + 1]$, the estimate of the reproduction number at time $T$ is not real-time anymore since it will be computed based on data up to time $T$ and the "future" data value available at time point $T+ 1$. For EpiEstim, the real-time characteristic of the $\mathcal{R}_t$ estimate is retained for any time point $t$ and is therefore more suitable for timely estimation. The real-time properties of EpiLPS and EpiEstim are compared and illustrated in Fig 8.

The extensive simulation results provided here, suggest that EpiLPS imposes itself as a robust retrospective estimation method. In particular, it seriously addresses a challenge faced by many existing methods, namely that $\mathcal{R}_t$ estimates typically lead or lag the true value [2]. EpiLPS is therefore a powerful retrospective tool to estimate the reproduction number during and/or after epidemic outbreaks. It is however less preferable than EpiEstim for real-time estimation and should therefore be used with care for timely purposes.

## Computing time and sensitivity analyses

The computational time of the EpiLPS algorithm is mainly affected by the number $K$ of B-splines specified in the basis and the total number of days $T$ of the epidemic. Table 5 gives an overview of the real elapsed time in seconds required to run the EpiLPS routines for different $(T, K)$ couples. Obviously, LPSMAP requires far less computational resources as it is a completely sampling-free approach relying on the MAP estimate of the hyperparameter vector. Even with an epidemic of roughly two months and $K = 60$, LPSMAP is extremely fast and delivers results in a fraction of a second. LPSMALA needs a larger computational budget as the algorithm relies on an iterative sampling scheme (MCMC). However, even for ($T = 60$,

**Table 5. Computational time (real elapsed time in seconds) of LPSMAP and LPSMALA (with a chain length of 3 000) for different combinations of $T$ (total number of days of the epidemic) and $K$ (total number of B-splines in the basis).** EpiLPS algorithm running on an Intel Xeon E-2186M CPU @2.90GHz with 16Go RAM.

| Method | | $K = 20$ | $K = 30$ | $K = 40$ | $K = 50$ | $K = 60$ |
|---|---|---|---|---|---|---|
| LPSMAP | $T = 20$ | 0.122 | 0.105 | 0.140 | 0.188 | 0.253 |
| | $T = 30$ | 0.091 | 0.124 | 0.185 | 0.242 | 0.326 |
| | $T = 40$ | 0.096 | 0.135 | 0.201 | 0.255 | 0.337 |
| | $T = 50$ | 0.083 | 0.109 | 0.156 | 0.200 | 0.276 |
| | $T = 60$ | 0.074 | 0.110 | 0.148 | 0.181 | 0.287 |
| LPSMALA | $T = 20$ | 3.098 | 3.499 | 4.151 | 4.832 | 5.886 |
| | $T = 30$ | 3.653 | 4.043 | 4.776 | 5.505 | 6.548 |
| | $T = 40$ | 4.167 | 4.663 | 5.425 | 6.126 | 7.238 |
| | $T = 50$ | 5.061 | 5.545 | 6.253 | 7.151 | 8.108 |
| | $T = 60$ | 5.913 | 6.362 | 7.151 | 8.062 | 9.074 |

$K = 60$), LPSMALA requires less than 10 seconds, which is a relatively reasonable time given the number of parameters involved in the model.

We assessed the sensitivity of the EpiLPS estimated reproduction number with respect to model inputs that are free to choose in order to check whether EpiLPS is robust with respect to different parameter choices. In particular, we focus on the sensitivity of the $\mathcal{R}_t$ fit (with LPSMAP) to the number $K$ of B-splines and to the parameters $a_\delta$ and $b_\delta$ of the Gamma hyper-prior on $\delta$. The sensitivity analyses are implemented in S2 Appendix and reveal a negligible sensitivity of the estimated $\mathcal{R}_t$ curve with respect to the above-mentioned parameters. We also discuss the sensitivity of the reproduction number estimates when computed over time domains of increasing width, for instance on $[1, T_1]$ and $[1, T_2]$ with $T_2 > T_1$. This gives an idea of the magnitude of variation in the estimated $\mathcal{R}_t$ in the domain $[1, T_1]$ when EpiLPS is actually fitted on the wider domain $[1, T_2]$. Results show that despite having values of $\mathcal{R}_t$ that vary (in the past) when applied to larger time domains due to the global smoothing approach inherent in EpiLPS, the estimated values of the reproduction number remain reasonably close to the target. S3 Appendix provides ancillary results on the estimation performance of the overdispersion parameter $\rho$ and sensitvity analyses of the computed credibles intervals for $\mathcal{R}_t$ with respect to different couples $(a_\delta, b_\delta)$.

### Application to observed case counts in infectious disease epidemics

**Epidemics of SARS-CoV-1 and influenza A H1N1.** In this section, the LPSMALA algorithm is applied on two historical outbreak datasets presented in [3]. In particular, we consider the 2003 SARS outbreak in Hong Kong and the 2009 pandemic influenza in a school in Pennsylvania (USA). We use $K = 40$ B-splines with a second-order penalty and the serial interval distributions provided in the EpiEstim package [4]. The LPSMALA algorithm is implemented with a chain of length 25 000. Acceptance rates for the generated chains are close to the optimal value of 57% and the posterior samples have converged according to the Geweke (1992) [42] diagnostic test (at the 1% level of significance). Fig 9 shows the smoothed epidemic curves and the estimated $\mathcal{R}_t$ for the two outbreaks. Results for the SARS data show that the reproduction number reaches a first peak during the third week, where $\hat{\mathcal{R}}_t = 9.67$ (95% CI: 5.19–16.47) and a second more moderate peak around week 6 with $\hat{\mathcal{R}}_t = 2.78$ (95% CI: 1.82–3.82). After day $t = 43$, the epidemic is under control and $\mathcal{R}_t$ smoothly decays below 1. For the pandemic influenza in Pennsylvania, in the end of the second week $\mathcal{R}_t$ is around 2.05 (95% CI: 1.21–3.06). During the middle of the third week, the situation is less severe and $\mathcal{R}_t$ points below 1.

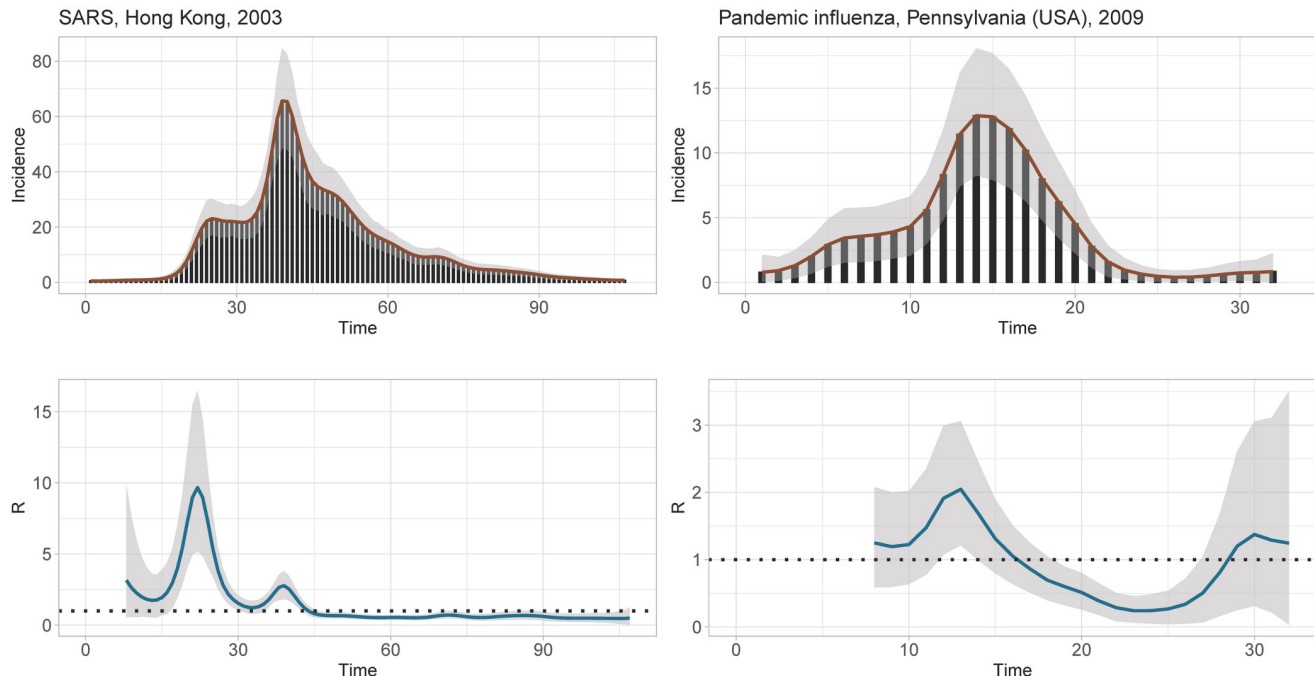

**Fig 9.** (Left column) EpiLPS fit for the epidemic curve (top) and the instantaneous reproduction number $\mathcal{R}_t$ (bottom) of the SARS outbreak in Hong Kong, 2003. (Right column) EpiLPS fit for the epidemic curve (top) and the instantaneous reproduction number $\mathcal{R}_t$ (bottom) of the pandemic influenza in Pennsylvania, 2009. The shaded area corresponds to the 95% credible interval at each day.

As noted in [3], a few cases appeared in the last days of the epidemic generating an upward trend in $\mathcal{R}_t$ estimates.

**Application on the SARS-CoV-2 pandemic.** The EpiLPS methodology is illustrated on the SARS-CoV-2 pandemic using publicly available data from the Covid-19 Data Hub [43] and its associated COVID19 package on CRAN (https://cran.r-project.org/package= COVID19). Country-level data on hospitalizations for Belgium, Denmark, Portugal and France from April 5th, 2020 to October 31st, 2021 is used and a serial interval distribution with a mean of 3 days (and standard deviation of 2.48 days) is assumed [44] discretized as $\varphi =$ {0.344, 0.316, 0.168, 0.104, 0.068}. In Fig 10, the estimated reproduction number obtained with EpiLPS and EpiEstim respectively, is shown for the four countries. Results are obtained with the LPSMAP algorithm using $K = 30$ B-splines and a second-order penalty. The gray shaded surface corresponds to 95% (pointwise) credible intervals for $\mathcal{R}_t$ with LPSMAP and the dashed curves are for EpiEstim. From a computational perspective, it takes less than 3 seconds to fit the EpiLPS model for the four countries. The fitted reproduction numbers reflect the different waves of the COVID-19 pandemic and the rise in infections in the beginning of September 2021. We also see that EpiLPS tends to follow the same trend as the estimates provided by EpiEstim, the only difference is that LPSMAP estimates appear globally smoother with credible intervals that are less wide for Belgium, Denmark and Portugal.

## Discussion

EpiLPS (an acronym for **Epi**demiological modeling with **L**aplacian-**P-S**plines) is a fast and flexible tool for Bayesian estimation of the instantaneous reproduction number $\mathcal{R}_t$ during epidemic outbreaks. The tool is flexible in the sense that (penalized) spline based approximations provide smoothed estimates of $\mathcal{R}_t$ with little computational effort and without the constraint

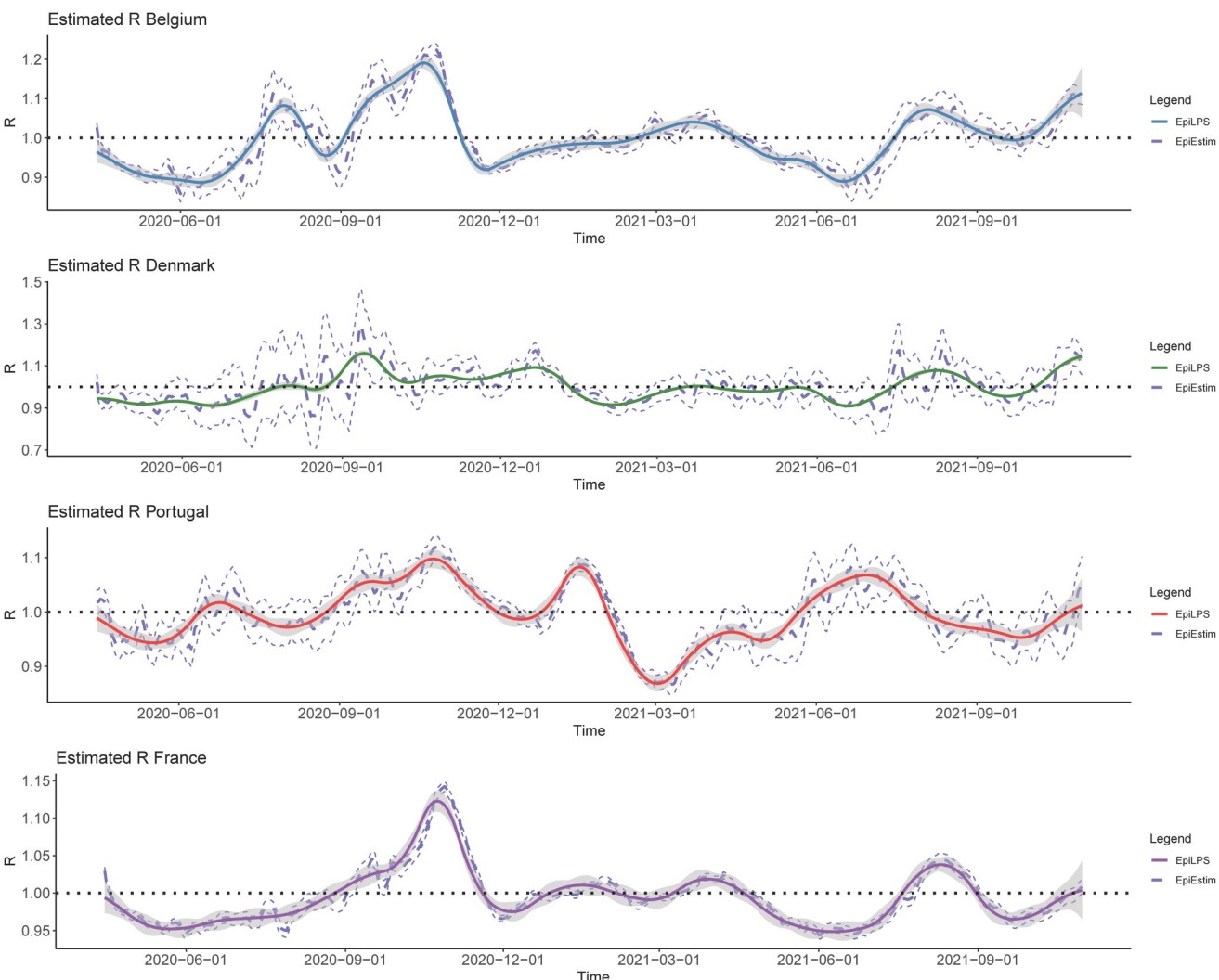

**Fig 10. Estimated reproduction number from 2020–04–05 to 2021–10–31 for Belgium, Denmark, Portugal and France with LPSMAP using *K* = 30 B-splines and a second-order penalty.** The shaded area corresponds to the 95% credible interval at each day. Dashed curves are results obtained with EpiEstim (with weekly sliding windows and estimated $\mathcal{R}_t$ reported at the end of the window).

of imposing any sliding window assumption that could potentially affect the timing and accuracy of the estimator. Moreover, the end user has the choice between a fully sampling-free approach (LPSMAP) or an efficient MCMC gradient-based approach with Langevin diffusions (LPSMALA) for inference. The available EpiLPS package (https://cran.r-project.org/package=EpiLPS) allows public health policy makers to analyze incoming data faster than existing methods relying on classic MCMC samplers, thus permitting them to be better informed when taking decisions on control measures for infectious disease outbreaks. Simulation studies in this paper provide encouraging results and support EpiLPS as being a robust tool capable of a precise tracking of $\mathcal{R}_t$ over time. The EpiLPS software package and the early website version (https://epilps.com) provide additional guiding material about the proposed methodology.

EpiLPS cannot be termed a real-time method in the same sense as in the Cori method and is therefore less preferred than EpiEstim for real-time analysis. Conceptually, EpiLPS and EpiEstim both use data from the past (EpiLPS also uses data from the future) to estimate the

instantaneous reproduction number, but the mechanisms underlying the use of past observations differ. The method of Cori looks back in time only as far as the width of the chosen time window in terms of infected individuals. EpiLPS on the contrary has a stronger reach as the P-splines smoother approximates the reproduction number globally (or blockwise), over the entire domain of the epidemic curve, i.e. retrospectively and also including future values (except for the estimate of $\mathcal{R}_t$ at the last day of the domain of the epidemic curve which makes use of the current day value and past values). This difference has important consequences and implies advantages as well as disadvantages. The advantage of working with a time window option as in EpiEstim is that one can control how far back in time to look in order to compute the desired $\mathcal{R}_t$ estimate. This is not an option in EpiLPS as the penalty parameter, the key driver of the degree of smoothness of the fitted $\mathcal{R}_t$ curve, is estimated within the model and is not fixed by the user. There is however no free lunch and the downside of having a time window choice in EpiEstim implies to face a trade-off between potential oversmoothing (with a wide time window) and undersmoothing (with a narrow time window). This trade-off is virtually absent in the EpiLPS setting as P-splines internally deal with the smoothing problem.

It is evident that when applying EpiLPS sequentially over time on epidemic curves with wider and wider domain length such as $[1, T_1], [1, T_2], [1, T_3]$ with $1 < T_1 < T_2 < T_3$, the $\mathcal{R}_t$ estimate over past days (for instance $t \in [1, T_1]$) will inevitably change as EpiLPS is by nature a global smoother. This past variability should not be seen as a drawback as it is essentially an "update" taking into account the fact that the method works with an epidemic curve with a longer domain. The real question is whether the past variability of the $\mathcal{R}_t$ estimate remains in a close neighborhood of the "true" value of the reproduction number for past days. On that side, the complete simulation study is rather convincing as it shows that EpiLPS is an accurate method that is successful in capturing the evolution of $\mathcal{R}_t$ over time.

There are also other aspects with respect to which EpiEstim and EpiLPS differ. For instance, prior specification in EpiEstim assumes a Gamma distributed prior on the reproduction number which is conjugate to the Poisson likelihood (EpiEstim assumes that incidence at time step $t$ is Poisson distributed), so that the posterior of $\mathcal{R}_t$ also has a Gamma distribution. In EpiLPS, the prior(s) are not directly imposed on the reproduction number, but on the spline parameters (and hyperparameters) and the resulting posterior distribution of $\mathcal{R}_t$ with LPSMAP is approximated by a lognormal distribution. Regarding computational complexity, EpiEstim and LPSMAP deliver estimates almost instantly, while LPSMALA requires a larger computing budget as it is a MCMC algorithm. We therefore recommend using LPSMALA over shorter epidemic durations and LPSMAP on longer outbreaks over several months. Our analysis suggests that EpiLPS might be more accurate than EpiEstim in presence of overdispersed epidemiological data, especially when it comes to quantify the uncertainty of $\mathcal{R}_t$ as EpiLPS is shown to have narrower credible intervals with good coverage performance. A main limitation is that EpiLPS is more prone to numerical instability (e.g. during hyperparameter optimization or in the Newton-Raphson algorithm for the Laplace approximation) than EpiEstim, although such problems were not encountered here. Finally, it is also worth mentioning that $\mathcal{R}_t$ estimates delivered by EpiLPS (and EpiEstim) are prone to potential biasing effects [2, 45] since the serial interval is used as a surrogate for the generation interval (time elapsed between infection events of an infector and an infectee) as the latter is less easily observed.

The EpiLPS project opens up several future research directions. A possible extension would be to formulate the EpiLPS model within a zero-inflated (Poisson) framework to cope with incidence time series characterized by an excess of zero counts. Another interesting extension would be to adapt the model to allow for regional variation and imported cases. Moreover, akin to EpiEstim, the EpiLPS methodology could be further developed to explicitly account for

uncertainty in the serial interval distribution. Finally, in face of long-lasting epidemic scenarios involving several variants characterized by different levels of virulence, it would be useful to extend the EpiLPS methodology to allow for smooth transitions of the estimated reproduction number accompanying the evolution of variants.

## Supporting information

**S1 Appendix. Details for the LPSMALA algorithm.** Analytical gradient for the Langevin-Hastings proposal and analytical version of the ratio of proposal distributions in the LPSMALA algorithm.
(PDF)

**S2 Appendix. Simulation results and computational time.** Complete simulation results (for EpiLPS and EpiEstim) when EpiEstim reports $\mathcal{R}_t$ at the window boundary, sensitivity analyses and computational time of EpiLPS.
(PDF)

**S3 Appendix. Further simulation and sensitivity results.** Complete simulation results (for EpiLPS and EpiEstim) when EpiEstim reports $\mathcal{R}_t$ at the window midpoint and additional sensitivity analyses.
(PDF)

## Author Contributions

**Conceptualization:** Oswaldo Gressani.

**Formal analysis:** Oswaldo Gressani.

**Funding acquisition:** Niel Hens, Christel Faes.

**Methodology:** Oswaldo Gressani.

**Software:** Oswaldo Gressani.

**Supervision:** Niel Hens.

**Validation:** Oswaldo Gressani.

**Visualization:** Oswaldo Gressani.

**Writing – original draft:** Oswaldo Gressani.

**Writing – review & editing:** Jacco Wallinga, Christian L. Althaus, Niel Hens, Christel Faes.

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
