## [Decision Letter · Decision Letter 0]

10 Mar 2022

Dear Dr Gressani,

Thank you very much for submitting your manuscript "EpiLPS: a fast and flexible Bayesian tool for near real-time estimation of the time-varying reproduction number" for consideration at PLOS Computational Biology.

As with all papers reviewed by the journal, your manuscript was reviewed by members of the editorial board and by several independent reviewers. In light of the reviews (below this email), we would like to invite the resubmission of a significantly-revised version that takes into account the reviewers' comments.

We cannot make any decision about publication until we have seen the revised manuscript and your response to the reviewers' comments. Your revised manuscript is also likely to be sent to reviewers for further evaluation.

Sincerely,

Claudio José Struchiner, M.D., Sc.D.

Associate Editor

PLOS Computational Biology

Tom Britton

Deputy Editor

PLOS Computational Biology

Reviewer's Responses to Questions

**Comments to the Authors:**

Reviewer #1: In this study, the authors propose a new method and R package to estimate the time-varying epidemic reproduction number. They use a simulation study to evaluate the performance of their method and compare it to an existing approach (EpiEstim), and present estimated reproduction numbers for historical SARS-CoV-1 and influenza epidemics as well as for SARS-Cov-2 in four European countries. I think the approach is interesting, particularly because it seems to allow for superspreading which few currently available methods do, but I have a number of queries that I would like to see addressed. A caveat: I haven’t had time to look into all the mathematical formulas in detail but for those I have looked into in depth they were all correct.

Major comments

Added value of the work

I think in the introduction and throughout the manuscript you could make it clearer from early on 1) why splines are useful (what are they trying to achieve in plain language compared to non spline based approaches) and 2) what is the unique aspect that your approach tackles (I think the use of a negative binomial likelihood, which I think you should emphasize more and again explain in plain English why it’s a useful feature, e.g. to model superspreading). I have to say I was very excited by the prospect of this paper precisely because I was hoping it would address gaps in the literature on estimating the reproduction number in the presence of super-spreading; but this doesn’t seem to be specifically addressed in the end, particularly in the simulation study which used a Poisson offspring distribution (see further comments below). So in summary, could the authors more clearly explain what makes their method different / better than others and could they show in their simulation study examples illustrating this more clearly?

Simulation study

I thought the simulation study could be hugely improved to better demonstrate the contexts in which the method performs well / better than others.

- First, why did you simulate from a Poisson and not a Negative binomial (see my comment above)? This seems very bizarre since I think this may be the main strength of your approach. I would suggest you expand your simulation study to demonstrate how well the method performs at various levels of superspreading / overdispersion. I would anticipate that benefits over for example EpiEstim may appear more clearly in the presence of overdispersion, which is not accounted for in most other approaches for estimation of R.

- Second, how did you choose the serial interval for each scenario? It seems weird to have different (rather ad hoc) ones for each scenario as it’s then hard to tell whether differences between scenarios come from the Rt profile or the change in SI. I would suggest using a couple of SI (maybe matching diseases e.g. flu and ebola for a short and longer SI?) and considering both SIs for each scenario.

- Third, how did you choose the values of K = 20 and second order penalty for the simulation study? Please consider alternative values (as you do with the window length for EpiEstim); I anticipate results may be quite sensitive to this choice.

- Finally, for EpiEstim results, how did you interpret these? i.e. for the 7 day time window, what “true” value of Rt (used for simulation) did you compare the 7 day average for? Rt at the window midpoint? The start? The end? It is very suspicious that in all scenarios (Figures 1-4, but maybe clearest Figure 3 the EpiEstim estimates seem to be “delayed”, as if you are not comparing like to like. And what priors did you use for EpiEstim? Also, what results would you obtain if you used 1 day windows for EpiEstim?

Potential limitations of the work

- The authors claim that “alleviating the window size assumption (as in EpiLPS) can be of interest” but they introduce a number of other parameters, including K, the number of spline knots, the choice of which is I think as arbitrary as that of the time window in EpiEstim albeit less easy to interpret. The authors should present sensitivity of their results to these parameters and add some discussion.

- Ability to detect sudden changes in Rt over time: “We assume that μ(t) evolves smoothly over the time course of the epidemic” � Does this mean the method would not be very appropriate to estimate rapid changes in R(t) for example the reduction in R associated with a lockdown or a similar sudden control measure? This is what your simulation study suggests, and should be added to the discussion, along with more discussion on the contexts in which the authors expect their method to be particularly useful.

- In figure 1 your approach seems able to detect a drop in Rt before it actually happens � how is that possible? This suggests to me that estimates of Rt will be informed by data from later (after t), which would potentially preclude real time use. Could the authors clarify and discuss this.

- What is the effect of the different priors and hyperpriors?

- A lot of the methodology is based on the Laplace approximation and I would have liked to see some evidence and/or discussion about the extent to which this approximation is going to be valid in the context of the renewal equation.

- Generally more discussion should be added on the caveats / subjective choices, and which types of epidemic contexts do we think this method will perform well / better than other methods.

Methodological clarifications

In section 2.4 I don’t think rho (the overdispersion parameter) appears at all. Could you please clarify the dependency on rho in this section?

Section 2.4.1, can you state more clearly what phi1, etc are; I assume phi1 = P(SI = 1)? Please define k more clearly. You should also highlight an assumption which is that p(SI <= 0) = 0 = p(SI > k). Please also more clarify in equation (9) whether y_t is the incidence of infections or symptomatic infections. In equation 9, please clearly state the values of s over which the sum is taken

Does the method lead to estimating the overdispersion parameter rho? If so, what estimates did you get for SARS-CoV-1 and 2 and flu and how do they compare with estimates from the literature obtained from analysing contact tracing data for example?

Related to my point about sensitivity to the number of spline knots K, I wonder how robust rho estimation would be to different choices for K, as I suspect it is hard to disentangle variations of Rt in time (the magnitude of which will be driven by K) from overdispersion (rho). I am not entirely sure these two are identifiable and would have hoped the authors show more clearly if this is true or not.

Comparison between your two methods and with EpiEstim

It is not clear how or when one should use LPSMALA versus LPSMAP; I would recommend that this is discussed further and that results of the two approaches are compared more systematically in figures. For example why did you use LPSMALA for SARS and Flu but LPSMAP for covid?

In Figure 5: how do these compare to the EpiEstim estimates? Cori et al. used 5 datasets but you only looked at 2 here, was there a specific reason? Could you include them all?

Implementation

Could the authors give some indication of computing time please.

Also, it is great that the R package is available on CRAN, but a shame that a vignette is not available. I would suggest that the authors use the SARS and flu examples for example as a basis for a vignette which would allow users to get started (I did notice some code is available to reproduce the results but the readme is not very informative so I think a vignette would be much better; I was using the link from the data availability section). The example in the epilps function runs well and fast (even with the LPSMALA method which is encouraging but I haven’t checked convergence was achieved).

Minor comments

Intro: I would avoid talking about “severity” of the outbreak in the context of the reproduction number as this may be confusing since R only measures transmissibility, not severity.

Equation 4: please clarify the notations, is L(theta, rho, D) the likelihood?

Do you really need to have both equations 12 and 13?

Reviewer #2: This paper presents a variant of the by now well known Cori et al (2013, 2021) approach to R_t estimation in an infectious disease outbreak based on smoothing the incidence count time series with splines.

The approach has the same basic problems as similar approaches to R_t, namely not accounting for cases not included in the time series of reported cases, delays in reporting, mixtures of differently infective strains, different incidences in different population groups.

Furthermore, a fixed serial time distribution is assumed (note that not all notified cases are due to symptom appearance and that recent research points to generation/serial times changing during an epidemic due to interventions and public behaviour...).

All this notwithstanding, R_t has turned out to be an important summary of population disease progress, used by decision makers, media and the public.

The paper is essentially concentrated on the computational part of R_t estimation as noted by the authors in the paragraph spanning pages 4-5, not on innovation of the concept of R_t itself. In fact, 14 pages out of 25 are essentially devoted to mathematical/statistical details of the used algorithms. Further 4 pages show the results of simulations of the proposed method compared to the EpiEstim method and finally there are 2 pages showing the results of the proposed new method applied to some real data.

Thus, the paper advocates use of the EpiLPS method as being slightly better than EpiEstim based on some simulations and with the theoretical advantage of not having to choose widths of moving windows in data. Perhaps, for a journal like PLoS Comp Biol, a much shorter paper with all the maths moved to an electronic appendix, would have been sufficient and quite adequate.

Some minor comments

On several occasions, R_t is called a "metric" in conflict with the usual mathematical meaning of the word.

In section 2.1, 3rd line, "infections" are called "contaminations", which may not be the most suitable synonym...

The four simulation scenarios are reasonable, but why are the serial interval distributions different?

Section 3.1 shows the proposed method seemingly having better adaptability to the "true" R_t but the comparison is with EpiEstim used with two fixed choices of window. Which choices of window widths have actually been advocated by EpiEstim users?

Reviewer #3: This is a very technical paper describing a method for estimating the reproduction ratio from epidemic data.

The method uses well known approaches (MCMC, laplace approximmation) that are combined and provide

an improvement in computation time.

1 - The method cannot be presented as "real-time". Real time should be reserved for methods

really attempting to estimate "in real time", that is characterized by

estimating R(t) from data up to time t. Here, as is common in such presentations, the authors

look at a global estimate obtained from the whole data. The R(t) estimate obtained from the renewal

equation is actually informed from the global spline smoothing.

2 - In the end, it is not very clear if the method should be prefered to the EpiEstim method. It is mentionned

that the coverage is better, but from what I read on EpiEstim they are using a window and estimating an average R(t) over the window,

with a corresponding confidence interval. So In the end, I'm not convinced that the comparison as presented here is fair,

because the confidence interval is for R(t) in your approach and for an average R(t) in the other case.

3 - There is one strange feature in the estimate seen in fig 2 for example : the R(t) starts to decrease before the date of intervention.

Do I have to tell you that this feature is always unwelcome? Indeed, this is exactly what will be used to say that "evidently" the intervention

was not responsible for the decrease in R(t) because "it started before the intervention". How could it be possible to change

the prior on the smoothing parameter automatically to change this?

4 - I suppose that running the method "in real time" during the course of an epidemic

would lead to changes in the estimates over the whole period because there would be changes in the smoothing parameter.

I would find it rather difficult to recommend the method if the "past" estimates would change as time passes.

5 - Are the confidence intervals "point-wise" or "trajectory-wise"?

6 - The method boasts using the NB formulation, but it is not shown how this makes the method more relevant. For example, was there

evidence in the examples that this was required? what was the difference between assuming NB or Poisson?

7 - I'm surprised that the daily pattern is not considered at all. In most countries, there was a strong such pattern. How comes

the spline does not try to adapt here to this pattern, since the prior seems to be favoring wiggliness?

8 - The model for using the NB does not really account for individual variation in transmission, where one would assume

that the offspring of each case is Xi = NB(R(t), rho). In this case, a more sensible description is i(t) = Sum Xi = NB(R(t) i(t-1), rho i(t-1))

assuming here for simplicity a generation time of one time unit. That is the dispersion parameter increases with incidence

This would be of interest to see whether you can accomodate this

description in here since it is much more relevant.

**Have the authors made all data and (if applicable) computational code underlying the findings in their manuscript fully available?**

Reviewer #1: Yes

Reviewer #2: Yes

Reviewer #3: **No: **although this is public data I am not sure it is available with the paper or in the package

PLOS authors have the option to publish the peer review history of their article (what does this mean?). If published, this will include your full peer review and any attached files.

Reviewer #1: No

Reviewer #2: No

Reviewer #3: No
---

## [Decision Letter · Decision Letter 1]

29 Jun 2022

Dear Dr Gressani,

Thank you very much for submitting your manuscript "EpiLPS: a fast and flexible Bayesian tool for estimation of the time-varying reproduction number" for consideration at PLOS Computational Biology.

As with all papers reviewed by the journal, your manuscript was reviewed by members of the editorial board and by several independent reviewers. In light of the reviews (below this email), we would like to invite the resubmission of a significantly-revised version that takes into account the reviewers' comments.

We cannot make any decision about publication until we have seen the revised manuscript and your response to the reviewers' comments. Your revised manuscript is also likely to be sent to reviewers for further evaluation.

Sincerely,

Claudio José Struchiner, M.D., Sc.D.

Associate Editor

PLOS Computational Biology

Tom Britton

Deputy Editor

PLOS Computational Biology

Reviewer's Responses to Questions

**Comments to the Authors:**

Reviewer #1: I appreciate the changes made by the authors, which I think have improved the manuscript, but I still have critical concerns about the work presented.

My three main concerns are as follows.

First, much of the paper is focused on a comparison with EpiEstim, but I think the comparison is not made in a fair way. Indeed Rt is compared to the Rt EpiEstim would estimate on the time window ]t-w;t] where w is the width of the estimation window. EpiEstim authors do explain this is the only way to have a fair evaluation of the real-time ability to recover Rt. But it does mean the estimates are “lagged”. If you are not tied by the real-time assumption, then a much better estimate of Rt is that obtained from EpiEstim on the time interval centered around t: [t – w/2; t + w/2]. This does require data after t but will no longer result in a lag. Since the method presented here is not applicable in real-time, and uses data after t, I think a much fairer baseline comparator for this new method would be this non lagged, non-real-time version of EpiEstim. I don’t think the article is publishable without showing this as I believe the performance of that version of EpiEstim will be much better. Hopefully the new methods developed by the authors will be even better (and looking at the figure results, I think they may well be) but this needs to be demonstrated. A related point re EpiEstim is that one of the apparent added value of the new method is that it can deal with overdispersion, unlike EpiEstim, however I could not see an obvious improvement of the method’s performance over EpiEstim’s in scenarios with more overdispersion. I suggest the authors comment on this. I would also like to have more details on how the performance indicators computed in table 2 are calculated – is this on average across the whole epidemic? Because it seems from figures as if EpiEstim has for example wider CrI in the early epidemic periods, but quite narrow in the late epidemics.

My second concern is that this is not applicable in real-time. Although the authors acknowledge it, they don’t really show the behaviour of the method would have if applied in real-time. Most of the recent applications of EpiEstim have been to track the transmissibility of SARS-CoV-2 in real-time, so the authors should 1) show how their method performs in real-time in an extended simulation study, or if not then more clearly that it SHOULD not be applied in real time and 2) if not aimed at real-time, more clearly highlight the contexts in which this would be a good method to use (e.g. for retrospective analyses of an epidemic).

At the moment without such practical application guidelines, this feels like a very theoretical exercise, and I don’t know to what extent it is useful.

My third worry is that although the theoretical foundations of this novel approach seem valid, it may feel a bit like a black box to the user. For example, the scaling of the covariance matrix is briefly mentioned in the main text but it’s unclear to me whether it’s up to the user to do this, and if so how. Without a vignette with more instructions on how to use this tool I wonder whether one could really use it in practice. Similarly, although sensitivity analyses are now presented to demonstrate the impact of changing the number of splines, I feel like the user is not given an easy rule of thumb that they could follow re what value they should choose.

A final comment is on the evaluation of the overdispersion parameter. Even though it is considered a nuisance parameter, and I understand it is not the primary objective of the method to estimate it, I would have liked to see how well it is re-estimated in the simulation study.

Overall, as currently written, I think it is unclear in what context this method may be useful, and if it would truly outperform EpiEstim if a fair comparison was undertaken, I therefore encourage the authors to further revise their manuscript to address these issues.

I also add some more minor comments below

P21 “calibrating different values of rho” should be “using different values of rho”?

End of p22: I struggle to understand how coverage can be so similar between two methods whilst one has much wider credible intervals; can you explain? Is this because one is really biased?

End of P24 you should also talk about the “price to pay” with the new method namely non applicability in real time

Figure 3 actually suggests more precise estimates of EpiEstim in that scenario?

Last sentence of section 3.1  I think this is really the main advantage of the new method and should be highlighted a bit more in the intro and conclusion; but the non-real-time nature of the approach should also be more emphasised

Figure 6: would be useful to see the underlying data (hospitalisations) and add some discussions about implications of using a proxy for infections (with alpha being more severe than previous variants R is going to be overestimated at the transition between the two when using hospitalisations).

First sentence of the discussion, you say “during epidemic outbreaks” but I think that is a push since the method won’t work in real-time. You should clarify what contexts and analyses this method will be useful for.

P31 “EpiLPS and EpiEstim both use data from the past to estimate the instantaneous reproduction number”  I think that is incorrect, i.e. EpiLPS also uses data from the future. If you want to demonstrate otherwise you need to present results of a real-time analysis.

P31 “The method of Cori looks back in time only as far as the width of the chosen time window”  That is also incorrect, it uses data from before that time window as well (it considers individuals infected in that time window but potential infectors from before as well as during that window).

P31 “over the entire domain of the epidemic curve”, i.e. including the future, I think this is important to highlight.

P31 “there is however no free lunch”  you should also highlight limitations of EpiLPS here namely the non real time nature which presumably greatly limit its applicability

P32, sentence starting “on that side”  again you haven’t shown that EpiLPS will work if you are trying to estimate Rt in real time which is a major limitation so please clarify here that it is an accurate method to capture the PAST evolution of Rt.

P32 when recommending to use LPSMALA for shorter epidemics, perhaps highlight that it performs better than LPSMAP?

P32 “EpiLPS might be more accurate than EpiEstim in presence of overdispersed epidemiological data”  what about in the absence of overdispersion? Again a detailed guide as to when one is best to use compared to the other would be really useful

Figure 11 supp2 and next ones: could you show how the CrI are affected by the changes in a_delta and b_delta parameters? This is important as it’s also one of the criteria you use to measure performance. Also in some of these figures the bias is actually quite a lot larger with low valus of a_delta and b_delta, which you should comment on in the matin text.

Table 4 supp2 please explicitly say which assumptions (e.g. time window of EpiEstim) you are using.

**Have the authors made all data and (if applicable) computational code underlying the findings in their manuscript fully available?**

Reviewer #1: Yes

Reviewer #2: Yes

PLOS authors have the option to publish the peer review history of their article (what does this mean?). If published, this will include your full peer review and any attached files.

Reviewer #1: No

Reviewer #2: No
---

## [Decision Letter · Decision Letter 2]

14 Sep 2022

Dear Dr Gressani,

Thank you very much for submitting your manuscript "EpiLPS: a fast and flexible Bayesian tool for estimation of the time-varying reproduction number" for consideration at PLOS Computational Biology. As with all papers reviewed by the journal, your manuscript was reviewed by members of the editorial board and by several independent reviewers. The reviewers appreciated the attention to an important topic. Based on the reviews, we are likely to accept this manuscript for publication, providing that you modify the manuscript according to the review recommendations.

Sincerely,

Claudio José Struchiner, M.D., Sc.D.

Academic Editor

PLOS Computational Biology

Tom Britton

Section Editor

PLOS Computational Biology

[LINK]

Reviewer's Responses to Questions

**Comments to the Authors:**

Reviewer #1: I thank the authors for taking the time to take on board my main comment from the last round of review. Although I am happy with these new result, I am afraid I still have some comments as detailed below.

Why present the new results in supplementary material? I think these are the main results and should be shown in the main text, whilst the results at the end of the interval currently in section 3.2 should be transferred to supplementary material.

The sensitivity analyses should also be altered to use this new window definition.

The main text results show that the best performance of EpiEstim in terms of nominal coverage is often obtained for a 1d window, therefore the authors should add this to their new results for completeness (both tables and figures in the current supplement 3), and so the reader can fully assess the comparison between EpiEstim with the 3 time windows and EpiLPS.

Adding figures for new sensitivity analyses on the github repository, with no explanation and no legend, is in my opinion not sufficient, these should be included as supplementary material and appropriately documented inthere.

Minor:

Page 11 what do you mean by “(in)directly”?

**Have the authors made all data and (if applicable) computational code underlying the findings in their manuscript fully available?**

Reviewer #1: Yes

PLOS authors have the option to publish the peer review history of their article (what does this mean?). If published, this will include your full peer review and any attached files.

Reviewer #1: No

Figure Files:

Data Requirements:

Reproducibility:

References:

---

## [Editor Report · Decision Letter 3]

30 Sep 2022

Dear Dr Gressani,

We are pleased to inform you that your manuscript 'EpiLPS: a fast and flexible Bayesian tool for estimation of the time-varying reproduction number' has been provisionally accepted for publication in PLOS Computational Biology.

Best regards,

Claudio José Struchiner, M.D., Sc.D.

Academic Editor

PLOS Computational Biology

Tom Britton

Section Editor

PLOS Computational Biology

---

## [Editor Report · Acceptance letter]

4 Oct 2022

PCOMPBIOL-D-22-00102R3 

EpiLPS: a fast and flexible Bayesian tool for estimation of the time-varying reproduction number

Dear Dr Gressani,

I am pleased to inform you that your manuscript has been formally accepted for publication in PLOS Computational Biology. Your manuscript is now with our production department and you will be notified of the publication date in due course.

With kind regards,

Zsofia Freund
